# Generalized Neural Sorting Networks with Error-Free Differentiable Swap Functions

**Jungtaek Kim**[1]     **Jeongbeen Yoon**[2]     **Minsu Cho**[2]
[1]University of Pittsburgh     [2]POSTECH
jungtaek.kim@pitt.edu     {jeongbeen,mscho}@postech.ac.kr

## Abstract

Sorting is a fundamental operation of all computer systems, having been a long-standing significant research topic. Beyond the problem formulation of traditional sorting algorithms, we consider sorting problems for more abstract yet expressive inputs, e.g., multi-digit images and image fragments, through a neural sorting network. To learn a mapping from a high-dimensional input to an ordinal variable, the differentiability of sorting networks needs to be guaranteed. In this paper we define a softening error by a differentiable swap function, and develop an error-free swap function that holds a non-decreasing condition and differentiability. Furthermore, a permutation-equivariant Transformer network with multi-head attention is adopted to capture dependency between given inputs and also leverage its model capacity with self-attention. Experiments on diverse sorting benchmarks show that our methods perform better than or comparable to baseline methods.

## 1 Introduction

Traditional sorting algorithms (Cormen et al., 2022), e.g., bubble sort, insertion sort, and quick sort, are a well-established approach to arranging given instances in computer science. Since such a sorting algorithm is a basic component to build diverse computer systems, it has been a long-standing significant research area in science and engineering. Moreover, sorting networks (Knuth, 1998; Ajtai et al., 1983), which are structurally designed as an abstract device with a fixed number of wires, have been widely used to perform a sorting algorithm on computing hardware, where each wire corresponds to a connection for a single swap operation.

Given an unordered sequence of $n$ elements $\mathbf{s} = [s_1, \ldots, s_n] \in \mathbb{R}^n$, the problem of sorting is defined to find a permutation matrix $\mathbf{P} \in \{0, 1\}^{n \times n}$ that transforms $\mathbf{s}$ into an ordered sequence $\mathbf{s}_{\mathrm{o}}$:

$$\mathbf{s}_{\mathrm{o}} = \mathbf{P}^\top \mathbf{s}, \tag{1}$$

where a sorting algorithm is a function $f$ of $\mathbf{s}$ that predicts a permutation matrix $\mathbf{P}$:

$$\mathbf{P} = f(\mathbf{s}). \tag{2}$$

We generalize the formulation of traditional sorting problems to handle more diverse and expressive types of inputs, e.g., multi-digit images and image fragments, which can contain ordinal information semantically. To this end, we extend the sequence of scalars $\mathbf{s}$ to the sequence of vectors $\mathbf{X} = [\mathbf{x}_1, \ldots, \mathbf{x}_n]^\top \in \mathbb{R}^{n \times d}$, where $d \gg 1$ is an input dimensionality, and consider the following:

$$\mathbf{X}_{\mathrm{o}} = \mathbf{P}^\top \mathbf{X}, \tag{3}$$

where $\mathbf{X}_{\mathrm{o}}$ and $\mathbf{X}$ are ordered and unordered inputs, respectively. This generalized sorting problem can be reduced to (1) if we are given a proper mapping $g$ from an input $\mathbf{x} \in \mathbb{R}^d$ to an ordinal value $s \in \mathbb{R}$. Without such a mapping $g$, predicting $\mathbf{P}$ in (3) remains more challenging than in (1) because $\mathbf{x}$ is often a highly implicative high-dimensional input. We address this generalized sorting problem by learning a neural sorting network together with a mapping $g$ in an end-to-end manner, given training data $\{(\mathbf{X}^{(i)}, \mathbf{P}_{\mathrm{gt}}^{(i)})\}_{i=1}^N$. The main challenge is to make the whole network $f([g(\mathbf{x}_1), \ldots, g(\mathbf{x}_n)])$ with mapping and sorting components differentiable in order to effectively train the network with a gradient-based learning scheme, which is not the case in general. To tackle

the differentiability issue for such a composite function, there has been recent research (Grover et al., 2019; Cuturi et al., 2019; Blondel et al., 2020; Petersen et al., 2021; 2022).

In this paper, following a sorting network-based sorting algorithm with differentiable swap functions (DSFs) (Petersen et al., 2021; 2022), we first define a softening error by a sorting network, which indicates a difference between original and smoothed elements. Then, we propose an error-free DSF that resolves an error accumulation problem induced by a soft DSF; this allows us to guarantee a zero error in mapping $\mathbf{X}$ to proper ordinal values. Based on this, we develop the sorting network with error-free DSFs where we adopt a permutation-equivariant Transformer architecture with multi-head attention (Vaswani et al., 2017) to capture dependency between high-dimensional inputs and also leverage the model capacity of the neural network with a self-attention scheme.

Our contributions can be summarized as follows: (i) We define a softening error that measures a difference between original and smoothed values; (ii) We propose an error-free DSF that resolves the error accumulation problem of conventional DSFs and is still differentiable; (iii) We adopt a permutation-equivariant network with multi-head attention as a mapping from inputs to ordinal variables $g(\mathbf{X})$, unlike $g(\mathbf{x})$; (iv) We demonstrate that our proposed methods are effective in diverse sorting benchmarks, compared to existing baseline methods.

## 2 SORTING NETWORKS WITH DIFFERENTIABLE SWAP FUNCTIONS

Following traditional sorting algorithms such as bubble sort, quick sort, and merge sort (Cormen et al., 2022) and sorting networks that are constructed by a fixed number of wires (Knuth, 1998; Ajtai et al., 1983), a swap function is a key ingredient of sorting algorithms and sorting networks:

$$(x', y') = \text{swap}(x, y), \tag{4}$$

where $x' = \min(x, y)$ and $y' = \max(x, y)$, which makes the order of $x$ and $y$ correct. For example, if $x > y$, then $x' = y$ and $y' = x$. Without loss of generality, we can express $\min(\cdot, \cdot)$ and $\max(\cdot, \cdot)$ with the following equations:

$$\min(x, y) = x\lfloor \sigma(y - x) \rceil + y\lfloor \sigma(x - y) \rceil \quad \text{and} \quad \max(x, y) = x\lfloor \sigma(x - y) \rceil + y\lfloor \sigma(y - x) \rceil, \tag{5}$$

where $\lfloor \cdot \rceil$ rounds to the nearest integer and $\sigma(\cdot) \in [0, 1]$ transforms an input to a bounded value, i.e., a probability over inputs. Computing (5) is straightforward, but they are not differentiable. To enable us to differentiate a swap function, the soft versions of $\min$ and $\max$ can be defined:

$$\overline{\min}(x, y) = x\sigma(y - x) + y\sigma(x - y) \quad \text{and} \quad \overline{\max}(x, y) = x\sigma(x - y) + y\sigma(y - x), \tag{6}$$

where $\sigma(\cdot)$ is differentiable. In addition to its differentiability, either (5) or (6) can be achieved with a sigmoid function $\sigma(x)$, i.e., a $s$-shaped function, which satisfies the following properties that (i) $\sigma(x)$ is non-decreasing, (ii) $\sigma(x) = 1$ if $x \to \infty$, (iii) $\sigma(x) = 0$ if $x \to -\infty$, (iv) $\sigma(0) = 0.5$, and (v) $\sigma(x) = 1 - \sigma(-x)$. Also, as discussed by Petersen et al. (2022), the choice of $\sigma$ affects the performance of neural network-based sorting network in theory as well as in practice. For example, an optimal monotonic sigmoid function, which is visualized in Figure 4, is defined as the following:

$$\sigma_{\mathcal{O}}(x) = \begin{cases} -\frac{1}{16}(\beta x)^{-1} & \text{if } \beta x < -0.25, \\ 1 - \frac{1}{16}(\beta x)^{-1} & \text{if } \beta x > 0.25, \\ \beta x + 0.5 & \text{otherwise,} \end{cases} \tag{7}$$

where $\beta$ is steepness; see the work (Petersen et al., 2022) for the details of these numerical and theoretical analyses. Here, we would like to emphasize that the important point of such monotonic sigmoid functions is strict monotonicity. However, as will be discussed in Section 3, it induces an error accumulation problem, which can degrade the performance of the sorting network.

By either (5) or (6), the permutation matrix $\mathbf{P}$ (henceforth, denoted as $\mathbf{P}_{\text{hard}}$ and $\mathbf{P}_{\text{soft}}$ for (5) and (6), respectively) is calculated by the following procedure of a sorting network: (i) Building a pre-defined sorting network with a fixed number of wires – a wire is a component for comparing and swapping two elements; (ii) Feeding an unordered sequence $\mathbf{s}$ into the pre-defined sorting network and calculating a wire-wise permutation matrix $\mathbf{P}_i$ for each wire $i$ iteratively; (iii) Calculating the permutation matrix $\mathbf{P}$ by multiplying all wire-wise permutation matrices.

As shown in Figure 1, a set of wires represents a set of swap operations that are operated simultaneously, so that each set produces an intermediate permutation matrix $\mathbf{P}_i$ at the $i$th step. Consequently,

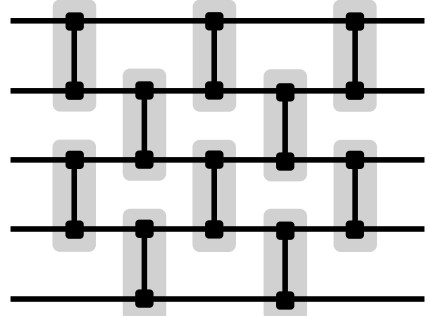

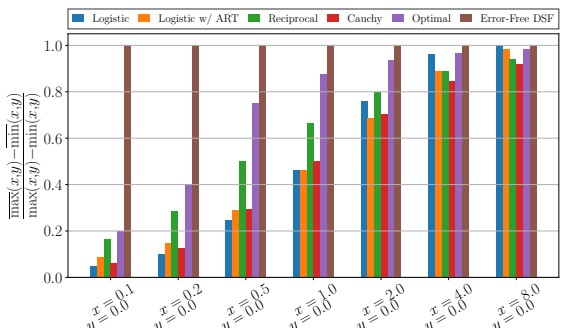

$$\mathbf{P}_1^\top \;\; \mathbf{P}_2^\top \;\; \mathbf{P}_3^\top \;\; \mathbf{P}_4^\top \;\; \mathbf{P}_5^\top = \mathbf{P}^\top$$

Figure 1: A sorting network with 5 wire sets and their permutation matrices.

Figure 2: Comparisons of diverse DSFs where a swap function is applied once. After a single operation, two input values $x$ and $y$ are softened while our error-free DSF does not change two values. If $|x - y|$ is small, softening will be more significant.

$\mathbf{P}^\top = \mathbf{P}_1^\top \mathbf{P}_2^\top \cdots \mathbf{P}_k^\top = (\mathbf{P}_k \cdots \mathbf{P}_2 \mathbf{P}_1)^\top$, where $k$ is the number of wire sets. For example, in Figure 1, $k = 5$.

The doubly-stochastic matrix property of $\mathbf{P}$ is shown by the following proposition:

**Proposition 1** (Modification of Lemma 3 in the work (Petersen et al., 2022))**.** *A permutation matrix* $\mathbf{P} \in \mathbb{R}^{n \times n}$ *is doubly-stochastic, which implies that* $\sum_{i=1}^{n} [\mathbf{P}]_{ij} = 1$ *and* $\sum_{j=1}^{n} [\mathbf{P}]_{ij} = 1$. *In particular, regardless of the definition of a swap function with* $\min$, $\max$, $\overline{\min}$, *and* $\overline{\max}$, *hard and soft permutation matrices, i.e.,* $\mathbf{P}_{\mathrm{hard}}$ *and* $\mathbf{P}_{\mathrm{soft}}$, *are doubly-stochastic.*

*Proof.* The proof of Proposition 1 is provided in Section D. □

In Sections 3 and 4, we present an error-free DSF and a neural sorting network with error-free DSFs.

## 3 ERROR-FREE DIFFERENTIABLE SWAP FUNCTIONS

Before introducing our error-free DSF, we start by describing the motivation of the error-free DSF.

Due to the nature of $\overline{\min}$ and $\overline{\max}$, which is described in (6), the monotonic DSF changes original input values. For example, if $x < y$, then $x < \overline{\min}(x, y)$ and $\overline{\max}(x, y) < y$ after applying the swap function. It can be a serious problem because changes by the DSF are accumulated as the DSF applies iteratively, called an error accumulation problem in this paper. The results of sigmoid functions such as the logistic, logistic with ART, reciprocal, Cauchy, and optimal monotonic functions, and also our error-free DSF are presented in Figure 2, where a swap function is applied once; see the work (Petersen et al., 2022) for the respective sigmoid functions. All DSFs except for our error-free DSF change two values, so that they can make two values not distinguishable. In particular, if a difference between two values is small, the consequence of softening is more significant than a case with a large difference. Moreover, if we apply a swap function repeatedly, they eventually become identical; see Figure 5 in Section B. While a swap function is not applied as many as it is tested in the synthetic example shown in Figure 5, it can still cause the error accumulation problem with a few operations. Here we formally define a softening error, which has been mentioned in this paragraph:

**Definition 1.** *Suppose that we are given* $x$ *and* $y$ *where* $x < y$. *By (6), these values* $x$ *and* $y$ *are softened by a monotonic DSF and they satisfy the following inequalities:*

$$x < x' = \overline{\min}(x, y) \le y' = \overline{\max}(x, y) < y. \tag{8}$$

*Therefore, we define a difference between the original and softened values,* $x' - x$ *or* $y - y'$:

$$y - y' = y - \overline{\max}(x, y) = x' - x = \overline{\min}(x, y) - x > 0, \tag{9}$$

*which is called a softening error in this paper. Without loss of generality, the softening error is* $\overline{\min}(x, y) - \min(x, y)$ *or* $\max(x, y) - \overline{\max}(x, y)$ *for any* $x, y$.

Note that (9) is satisfied by $y - \overline{\max}(x, y) = y(1 - \sigma(y - x)) - x\sigma(x - y) = y\sigma(x - y) - x(1 - \sigma(y - x)) = \overline{\min}(x, y) - x$, using (6) and $\sigma(x - y) = 1 - \sigma(y - x)$.

With Definition 1, we are able to specify the seriousness of the error accumulation problem:

**Proposition 2.** *Suppose that $x$ and $y$ are given and a DSF is applied $k$ times. Assuming an extreme scenario that $k \to \infty$, error accumulation becomes $(\max(x, y) - \min(x, y))/2$, under the assumption that $\nabla_x \sigma(x) > 0$.*

*Proof.* The proof of this proposition can be found in Section E. $\square$

As mentioned in the proof of Proposition 2 and empirically shown in Figure 2, a swap function with relatively large $\nabla_x \sigma(x)$ changes the original values $x, y$ significantly compared to a swap function with relatively small $\nabla_x \sigma(x)$ – they tend to become identical with the small number of operations in the case of large $\nabla_x \sigma(x)$.

In addition to the error accumulation problem, such a DSF depends on the scale of $|x - y|$ as shown in Figure 2. If $x < y$ but $x$ and $y$ are close enough, $\sigma(y - x)$ is between $0.5$ and $1$, which implies that the error can be induced by the scale of $|x - y|$ as well.

To tackle the aforementioned problem of error accumulation, we propose an error-free DSF:

$$(x', y') = \mathrm{swap}_{\text{error-free}}(x, y), \tag{10}$$

where

$$x' = \left(\min(x, y) - \overline{\min}(x, y)\right)_{\mathrm{sg}} + \overline{\min}(x, y) \quad \text{and} \quad y' = (\max(x, y) - \overline{\max}(x, y))_{\mathrm{sg}} + \overline{\max}(x, y). \tag{11}$$

Note that sg indicates that gradients are stopped amid backward propagation, inspired by a straight-through estimator (Bengio et al., 2013). At a step for forward propagation, the error-free DSF produces $x' = \min(x, y)$ and $y' = \max(x, y)$. On the contrary, at a step for backward propagation, the gradients of $\overline{\min}$ and $\overline{\max}$ are used to update learnable parameters. Consequently, our error-free DSF does not smooth the original elements as shown in Figure 2 and our DSF shows 100% accuracy for $\mathrm{acc}_{\mathrm{em}}$ and $\mathrm{acc}_{\mathrm{ew}}$ (see Section 5 for their definitions) as shown in Figure 6. Compared to our DSF, the existing DSFs do not correspond the original elements to the elements that have been compared and fail to achieve reasonable performance as a sequence length increases, in the cases of Figure 6.

By (5), (6), and (11), we obtain the following:

$$x' = ((x\lfloor \sigma(y - x)\rceil + y\lfloor \sigma(x - y)\rceil) - (x\sigma(y - x) + y\sigma(x - y)))_{\mathrm{sg}} + (x\sigma(y - x) + y\sigma(x - y))$$
$$= x \left((\lfloor \sigma(y - x)\rceil - \sigma(y - x))_{\mathrm{sg}} + \sigma(y - x)\right) + y \left((\lfloor \sigma(x - y)\rceil - \sigma(x - y))_{\mathrm{sg}} + \sigma(x - y)\right), \tag{12}$$
$$y' = x \left((\lfloor \sigma(x - y)\rceil - \sigma(x - y))_{\mathrm{sg}} + \sigma(x - y)\right) + y \left((\lfloor \sigma(y - x)\rceil - \sigma(y - x))_{\mathrm{sg}} + \sigma(y - x)\right), \tag{13}$$

which can be used to define a permutation matrix with the error-free DSF. For example, if $n = 2$, a permutation matrix $\mathbf{P}$ over $[x, y]$ is

$$\mathbf{P} = \begin{bmatrix} (\lfloor \sigma(y - x)\rceil - \sigma(y - x))_{\mathrm{sg}} + \sigma(y - x) & (\lfloor \sigma(x - y)\rceil - \sigma(x - y))_{\mathrm{sg}} + \sigma(x - y) \\ (\lfloor \sigma(x - y)\rceil - \sigma(x - y))_{\mathrm{sg}} + \sigma(x - y) & (\lfloor \sigma(y - x)\rceil - \sigma(y - x))_{\mathrm{sg}} + \sigma(y - x) \end{bmatrix}. \tag{14}$$

To sum up, we can describe the following proposition on our error-free DSF, $\mathrm{swap}_{\text{error-free}}(\cdot, \cdot)$:

**Proposition 3.** *By (11), the softening error $x' - \min(x, y)$ or $\max(x, y) - y'$ for an error-free DSF is zero.*

*Proof.* The proof of this proposition is presented in Section F. $\square$

## 4 NEURAL SORTING NETWORKS WITH ERROR-FREE DIFFERENTIABLE SWAP FUNCTIONS

In this section we propose a generalized neural network-based sorting network with an error-free DSF and a permutation-equivariant neural network, considering the properties covered in Section 3.

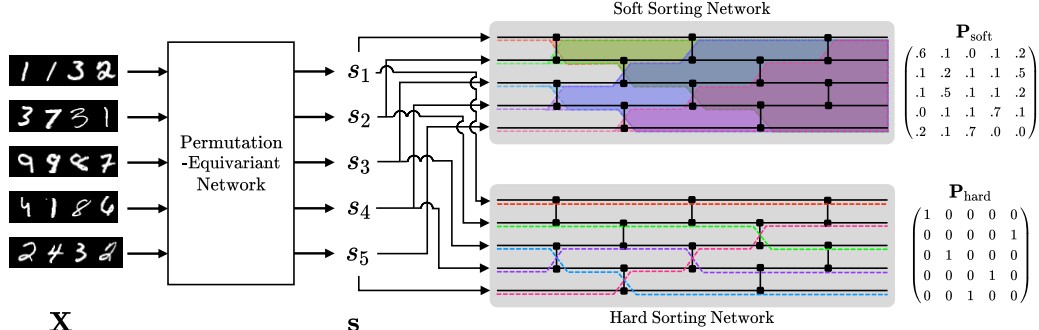

Figure 3: Illustration of our neural sorting network with error-free DSFs. Given high-dimensional inputs $\mathbf{X}$, a permutation-equivariant network produces a vector of ordinal variables $\mathbf{s}$, which is used to be swapped using a soft or hard sorting network.

First, we describe a procedure for transforming high-dimensional inputs to ordinal scores. Such a mapping $g : \mathbb{R}^d \to \mathbb{R}$, which consists of a set of learnable parameters, has to satisfy a permutation-equivariant property:

$$[g(\mathbf{x}_{\pi_1}), g(\mathbf{x}_{\pi_2}), \ldots, g(\mathbf{x}_{\pi_n})] = \pi([g(\mathbf{x}_1), g(\mathbf{x}_2), \ldots, g(\mathbf{x}_n)]), \qquad (15)$$

where $\pi_i = [\pi([1, 2, \ldots, n])]_i \; \forall i \in [n]$, for any permutation function $\pi$. Typically, an instance-wise neural network, which is applied to each element in a sequence given, is permutation-equivariant (Zaheer et al., 2017). Based on this property, instance-wise CNNs are employed in differentiable sorting algorithms (Grover et al., 2019; Cuturi et al., 2019; Petersen et al., 2021; 2022). However, such an instance-wise architecture is limited since it is ineffective for capturing essential features from a sequence. Some types of neural networks such as long short-term memory (Hochreiter & Schmidhuber, 1997) and the standard Transformer architecture (Vaswani et al., 2017) are capable of modeling a sequence of instances, utilizing recurrent connections, scaled dot-product attention, or parameter sharing across elements. While they are powerful for modeling a sequence, they are not obviously permutation-equivariant. Instead of such permutation-sensitive models, we adopt a robust Transformer-based network that satisfies the permutation-equivariant property, which is inspired by the recent work (Vaswani et al., 2017; Lee et al., 2019).

To explain our network, we briefly introduce scaled dot-product attention and multi-head attention:

$$\text{att}(\mathbf{Q}, \mathbf{K}, \mathbf{V}) = \text{softmax}\left(\frac{\mathbf{Q}\mathbf{K}^\top}{\sqrt{d_m}}\right)\mathbf{V} \quad \text{and} \quad \text{mha}(\mathbf{Q}, \mathbf{K}, \mathbf{V}) = [\text{head}_1, \text{head}_2, \ldots, \text{head}_h]\mathbf{W}_o, \tag{16}$$

where $\text{head}_i = \text{att}(\mathbf{Q}\mathbf{W}_q^{(i)}, \mathbf{K}\mathbf{W}_k^{(i)}, \mathbf{V}\mathbf{W}_v^{(i)})$, $\mathbf{Q}, \mathbf{K}, \mathbf{V} \in \mathbb{R}^{n \times hd_m}$, $\mathbf{W}_q^{(i)}, \mathbf{W}_k^{(i)}, \mathbf{W}_v^{(i)} \in \mathbb{R}^{hd_m \times d_m}$, and $\mathbf{W}_o \in \mathbb{R}^{hd_m \times hd_m}$. Similar to the Transformer network, a series of mha blocks is stacked with layer normalization (Ba et al., 2016) and residual connections (He et al., 2016), and in this paper $\mathbf{X}$ is processed by $\text{mha}(\mathbf{Z}, \mathbf{Z}, \mathbf{Z})$ where $\mathbf{Z} = g'(\mathbf{X})$ or $\mathbf{Z}$ is the output of a previous layer; see Section I for the details of the architectures. Note that $g'(\cdot)$ is an instance-wise embedding layer, e.g., a simple fully-connected network or a simple CNN. Importantly, compared to the standard Transformer model, our network does not include a positional embedding, in order to satisfy the permutation-equivariant property; $\text{mha}(\mathbf{Z}, \mathbf{Z}, \mathbf{Z})$ satisfies (15) for the permutation of $\mathbf{z}_1, \mathbf{z}_2, \ldots, \mathbf{z}_n$ where $\mathbf{Z} = [\mathbf{z}_1, \mathbf{z}_2, \ldots, \mathbf{z}_n]^\top$. The output of our network is $\mathbf{s}$, followed by the last instance-wise fully-connected layer. Finally, as shown in Figure 3, our sorting network is able to produce differentiable permutation matrices over $\mathbf{s}$, i.e., $\mathbf{P}_{\text{hard}}$ and $\mathbf{P}_{\text{soft}}$, by utilizing (11) and (6), respectively. Note that $\mathbf{P}_{\text{hard}}$ and $\mathbf{P}_{\text{soft}}$ are doubly-stochastic by Proposition 1. In addition, the details of the permutation-equivariant network with multi-head attention are briefly visualized in Figure 7.

To learn the permutation-equivariant network $g$, we define both objectives for $\mathbf{P}_{\text{soft}}$ and $\mathbf{P}_{\text{hard}}$:

$$\mathcal{L}_{\text{soft}} = -\sum_{i=1}^{n}\sum_{j=1}^{n}[\mathbf{P}_{\text{gt}}\log\mathbf{P}_{\text{soft}} + (1 - \mathbf{P}_{\text{gt}})\log(1 - \mathbf{P}_{\text{soft}})]_{ij}, \tag{17}$$

$$\mathcal{L}_{\text{hard}} = \|\mathbf{X}_{\text{o,hard}} - \mathbf{X}_{\text{o,gt}}\|_F^2 = \|\mathbf{P}_{\text{hard}}^\top\mathbf{X} - \mathbf{P}_{\text{gt}}^\top\mathbf{X}\|_F^2, \tag{18}$$

Table 1: Results on sorting the four-digit MNIST dataset. The results are measured in $\mathrm{acc_{em}}$ and $\mathrm{acc_{ew}}$ (in parentheses). FLOPs is on the basis of a sequence length 3. All the values are averaged over 5 runs with different seeds.

| Method | Model | Sequence Length | | | | | | FLOPs | #Param. |
|---|---|---|---|---|---|---|---|---|---|
| | | 3 | 5 | 7 | 9 | 15 | 32 | | |
| NeuralSort | | 91.9 (94.5) | 77.7 (90.1) | 61.0 (86.2) | 43.4 (82.4) | 9.7 (71.6) | 0.0 (38.8) | | |
| Sinkhorn Sort | | 92.8 (95.0) | 81.1 (91.7) | 65.6 (88.2) | 49.7 (84.7) | 12.6 (74.2) | 0.0 (41.2) | | |
| Fast Sort & Rank | | 90.6 (93.5) | 71.5 (87.2) | 49.7 (81.3) | 29.0 (75.2) | 2.8 (60.9) | – | | |
| Logistic | CNN | 92.0 (94.5) | 77.2 (89.8) | 54.8 (83.6) | 37.2 (79.4) | 4.7 (62.3) | 0.0 (56.3) | 130M | 855K |
| Logistic w/ ART | | 94.3 (96.1) | 83.4 (92.6) | 71.6 (90.0) | 56.3 (86.7) | 23.5 (79.4) | 0.5 (64.9) | | |
| Diffsort Reciprocal | | 94.4 (96.1) | 85.0 (93.3) | 73.4 (90.7) | 60.8 (88.1) | 30.2 (81.9) | 1.0 (66.8) | | |
| Cauchy | | 94.2 (96.0) | 84.9 (93.2) | 73.3 (90.5) | 63.8 (89.1) | 31.1 (82.2) | 0.8 (63.3) | | |
| Optimal | | 94.6 (96.3) | 85.0 (93.3) | 73.6 (90.7) | 62.2 (88.5) | 31.8 (82.3) | 1.4 (67.9) | | |
| | CNN | 95.2 (96.7) | 87.2 (94.2) | 76.6 (91.6) | 64.8 (89.2) | 34.7 (83.3) | 2.1 (69.2) | 130M | 855K |
| Ours Error-Free DSFs | Transformer-S | 95.9 (97.1) | 94.8 (97.5) | 90.8 (96.5) | 86.9 (95.7) | 74.3 (93.6) | 37.8 (87.7) | 130M | 665K |
| | Transformer-L | 96.5 (97.5) | 95.4 (97.7) | 92.9 (97.2) | 90.1 (96.5) | 82.5 (95.0) | 46.2 (88.9) | 137M | 3.104M |

where $\mathbf{P}_{\mathrm{gt}}$ is a ground-truth permutation matrix. Note that all the operations in $\mathcal{L}_{\mathrm{soft}}$ are entry-wise. Similar to (17), the objective (18) for $\mathbf{P}_{\mathrm{hard}}$ should be designed as the form of binary cross-entropy, which tends to be generally robust for training deep neural networks. However, we struggle to apply the binary cross-entropy for $\mathbf{P}_{\mathrm{hard}}$ into our problem formulation, due to discretized loss values. In particular, the form of cross-entropy for $\mathbf{P}_{\mathrm{hard}}$ can be used to train the sorting network, but degrades its performance in our preliminary experiments. Thus, we choose the objective for $\mathbf{P}_{\mathrm{hard}}$ as $\|\mathbf{X}_{\mathrm{o,hard}} - \mathbf{X}_{\mathrm{o,gt}}\|_F^2$ with the Frobenius norm, which helps to train the network more robustly.

In addition, using a proposition on splitting $\mathbf{P}_{\mathrm{hard}}$, which is discussed in Section H, the objective (18) for $\mathbf{P}_{\mathrm{hard}}$ can be modified by splitting $\mathbf{P}_{\mathrm{hard}}$, $\mathbf{P}_{\mathrm{gt}}$, and $\mathbf{X}$, which is able to reduce the number of possible permutations; see the associated section for details. Eventually, our network $g$ is trained by the combined loss $\mathcal{L} = \mathcal{L}_{\mathrm{soft}} + \lambda\mathcal{L}_{\mathrm{hard}}$, where $\lambda$ is a balancing hyperparameter; an analysis on $\lambda$ can be found in the appendices. As mentioned above, a landscape of $\mathcal{L}_{\mathrm{hard}}$ is not smooth due to the property of a straight-through estimator, even though we use $\mathcal{L}_{\mathrm{hard}}$. Thus, we combine both objectives to the form of a single loss, which is widely adopted in the deep learning community.

## 5 EXPERIMENTS

We demonstrate experimental results to show the validity of our methods. Our neural network-based sorting network aims to solve two benchmarks: sorting (i) multi-digit images and (ii) image fragments. Unless otherwise specified, an odd-even sorting network is used in the experiments. We measure the performance of each method in $\mathrm{acc_{em}}$ and $\mathrm{acc_{ew}}$:

$$\mathrm{acc_{em}} = \frac{\sum_{i=1}^{N} \bigcap_{j=1}^{n} \mathbf{1}\left(\left[\hat{\mathbf{s}}^{(i)}\right]_j = \left[\tilde{\mathbf{s}}^{(i)}\right]_j\right)}{N} \quad \text{and} \quad \mathrm{acc_{ew}} = \frac{\sum_{i=1}^{N} \sum_{j=1}^{n} \mathbf{1}\left(\left[\hat{\mathbf{s}}^{(i)}\right]_j = \left[\tilde{\mathbf{s}}^{(i)}\right]_j\right)}{Nn}, \tag{19}$$

where $\mathrm{argsort}$ returns indices to sort a given vector and $\mathbf{1}(\cdot)$ is an indicator function. Note that

$$\hat{\mathbf{s}}^{(i)} = \mathrm{argsort}\left(\mathbf{P}_{\mathrm{gt}}^{(i)\top}\mathbf{s}^{(i)}\right) \quad \text{and} \quad \tilde{\mathbf{s}}^{(i)} = \mathrm{argsort}\left(\mathbf{P}^{(i)\top}\mathbf{s}^{(i)}\right). \tag{20}$$

We attempt to match the capacities of the Transformer-based models to the conventional CNNs. As described in Tables 1, 2, and 3, the capacities of the Transformer-Small models are smaller than or similar to the capacities of the CNNs in terms of FLOPs and the number of parameters.

### 5.1 SORTING MULTI-DIGIT IMAGES

**Datasets.** As steadily utilized in the previous work (Grover et al., 2019; Cuturi et al., 2019; Blondel et al., 2020; Petersen et al., 2021; 2022), we create a four-digit dataset by concatenating four images from the MNIST dataset (LeCun et al., 1998); see Figure 3 for some examples of the dataset. On the other hand, the SVHN dataset (Netzer et al., 2011) contains multi-digit numbers extracted from street view images and is therefore suitable for sorting.

Table 2: Results on sorting the SVHN dataset. FLOPs is computed on the basis of a sequence length 3. All the values are averaged over 5 runs with different seeds.

| Method | Model | Sequence Length | | | | | FLOPs | #Param. |
|---|---|---|---|---|---|---|---|---|
| | | 3 | 5 | 7 | 9 | 15 | | |
| Diffsort | | Logistic | | | | | | |
| | | Logistic 76.3 (83.2) | 46.0 (72.7) | 21.8 (63.9) | 13.5 (61.7) | 0.3 (45.9) | | |
| | | Logistic w/ ART 83.2 (88.1) | 64.1 (82.1) | 43.8 (76.5) | 24.2 (69.6) | 2.4 (56.8) | | |
| | CNN | Reciprocal 85.7 (89.8) | 68.8 (84.2) | 53.3 (80.0) | 40.0 (76.3) | 13.2 (66.0) | 326M | 1.226M |
| | | Cauchy 85.5 (89.6) | 68.5 (84.1) | 52.9 (79.8) | 39.9 (75.8) | 13.7 (66.0) | | |
| | | Optimal 86.0 (90.0) | 67.5 (83.5) | 53.1 (80.0) | 39.1 (76.0) | 13.2 (66.3) | | |
| Ours | Error-Free DSFs | CNN 86.8 (90.6) | 68.9 (84.5) | 53.4 (80.4) | 40.0 (77.0) | 12.0 (65.3) | 326M | 1.226M |
| | | Transformer-S 86.6 (90.2) | 72.6 (85.7) | 62.5 (83.5) | 48.6 (79.3) | 19.3 (69.6) | 210M | 1.223M |
| | | Transformer-L 88.0 (91.2) | 74.0 (86.3) | 63.9 (83.8) | 50.2 (80.1) | 21.7 (71.2) | 332M | 3.475M |

Table 3: Results on sorting image fragments of MNIST and CIFAR-10. $2 \times 2$ and $3 \times 3$ indicate the numbers of fragments, and $14 \times 14$, $9 \times 9$, $16 \times 16$, and $10 \times 10$ (in parentheses) indicate the sizes of image fragments. FLOPs is computed on the basis of the MNIST $2 \times 2$ ($14 \times 14$) case and the CIFAR-10 $2 \times 2$ ($16 \times 16$) case. All the values are averaged over 5 runs with different seeds.

| Method | Model | MNIST | | | | CIFAR-10 | | | |
|---|---|---|---|---|---|---|---|---|---|
| | | $2 \times 2$ ($14 \times 14$) | $3 \times 3$ ($9 \times 9$) | FLOPs | #Param. | $2 \times 2$ ($16 \times 16$) | $3 \times 3$ ($10 \times 10$) | FLOPs | #Param. |
| Diffsort | CNN | Logistic 98.5 (99.0) | 5.3 (42.9) | | | 56.9 (73.6) | 0.8 (27.7) | | |
| | | Logistic w/ ART 98.4 (99.1) | 5.4 (42.9) | | | 56.7 (73.4) | 0.7 (27.7) | | |
| | | Reciprocal 98.4 (99.2) | 5.3 (42.9) | 1.498M | 84K | 56.7 (73.4) | 0.7 (27.8) | 1.663M | 85K |
| | | Cauchy 98.4 (99.2) | 5.3 (42.9) | | | 56.9 (73.4) | 0.9 (27.9) | | |
| | | Optimal 98.4 (99.1) | 5.3 (43.0) | | | 56.6 (73.4) | 0.7 (27.7) | | |
| Ours | Error-Free DSFs | CNN 98.4 (99.2) | 5.2 (42.6) | 1.498M | 84K | 56.9 (73.6) | 0.8 (28.0) | 1.663M | 85K |
| | | Transformer 98.6 (99.2) | 5.6 (43.7) | 946K | 87K | 58.1 (74.2) | 0.9 (28.3) | 1.111M | 87K |

**Experimental Details.** We conduct the experiments 5 times by varying random seeds to report the average of $\mathrm{acc_{em}}$ and $\mathrm{acc_{ew}}$, and use the optimal monotonic sigmoid function as DSFs. The performance of each model is measured by a test dataset. We use the AdamW optimizer (Loshchilov & Hutter, 2018), and train each model for 200,000 steps on the four-digit MNIST dataset and 300,000 steps on the SVHN dataset. Unless otherwise noted, we follow the same settings of the work (Petersen et al., 2022) for fair comparisons. Missing details are described in Section J.

**Results.** Tables 1 and 2 show the results of the previous work such as NeuralSort (Grover et al., 2019), Sinkhorn Sort (Cuturi et al., 2019), Fast Sort & Rank (Blondel et al., 2020), and Diffsort (Petersen et al., 2021; 2022), and our methods on the MNIST and SVHN datasets, respectively. When we use the conventional CNN as a permutation-equivariant network, our method shows better than or comparable to the previous methods. As we exploit more powerful models, i.e., the Transformer-Small and Transformer-Large permutation-equivariant models, our approaches show better results compared to other existing methods including our method with the conventional CNN.[1]

## 5.2 SORTING IMAGE FRAGMENTS

**Datasets.** For experiments on sorting image fragments, we use two datasets: the MNIST dataset (LeCun et al., 1998) and the CIFAR-10 dataset (Krizhevsky & Hinton, 2009). Similar to the work (Mena et al., 2018), we create multiple fragments or patches from a single-digit image of the MNIST dataset to utilize themselves as inputs – for example, 4 fragments of size $14 \times 14$ or 9 fragments of size $9 \times 9$ are created from a single image. Similarly, the image included in the CIFAR-10 dataset, which contains one of various objects, e.g., birds and cats, is split to multiple patches, and then is used to the experiments on sorting image fragments. See Table 3 for the details of the image fragments and their sizes.

---

[1]Thanks to many open-source projects, we can easily run the baseline methods. However, it is difficult to reproduce some results due to unknown random seeds. For this reason, we bring the results from the work (Petersen et al., 2022), and use fixed random seeds, i.e., 42, 84, 126, 168, 210, for our methods.

**Experimental Details.** Similar to the experiments on sorting multi-digit images, an optimal monotonic sigmoid function is used as DSFs. Since the size of inputs is much smaller than the experiments on sorting multi-digit images, shown in Section 5.1, we modify the architectures of the CNNs and the Transformer-based models. We reduce the kernel size of convolution layers from 5 to 3 and make strides 2. Due to the small input sizes, we omit the results by the Transformer-Large model for these experiments. Additionally, max-pooling operations are removed. Similar to the experiments in Section 5.1, we use the AdamW optimizer (Loshchilov & Hutter, 2018). Moreover, each model is trained for 50,000 steps when the number of fragments is $2 \times 2$, i.e., when a sequence length is 4, and 100,000 steps for $3 \times 3$ fragments, i.e., when a sequence length is 9. Additional information including the details of neural architectures can be found in Sections I and J.

**Results.** Table 3 represents the experimental results on both datasets of image fragments, which are created from the MNIST and CIFAR-10 datasets. Similar to the experiments on sorting multi-digit images, the more powerful architecture improves performance in this task.

According to the experimental results, we achieve satisfactory performance by applying the error-free DSFs, combined loss, and Transformer-based models with multi-head attention. We provide detailed discussion on how they contribute to the performance gains in Section 7, and empirical studies on steepness, learning rate, and a balancing hyperparameter in Section 7 and the appendices.

## 6 RELATED WORK

**Differentiable Sorting Algorithms.** To allow us to differentiate a sorting algorithm, Grover et al. (2019) have proposed the continuous relaxation of argsort operator, which is named NeuralSort. In this work, the output of NeuralSort only satisfies the row-stochastic matrix property, although Grover et al. (2019) attempt to employ a gradient-based optimization strategy in learning a neural sorting algorithm. Cuturi et al. (2019) propose a smoothed ranking and sorting operator using optimal transport, which is the natural relaxation for assignments. To reduce the cost of the optimal transport, the Sinkhorn algorithm (Cuturi, 2013) is used. Then, Blondel et al. (2020) have proposed a differentiable sorting and ranking operator with $\mathcal{O}(n \log n)$ time and $\mathcal{O}(n)$ space complexities, which is named Fast Rank & Sort, by constructing differentiable operators as projections on permutahedron. Petersen et al. (2021) have suggested a differentiable sorting network with relaxed conditional swap functions. Recently, the same authors analyze the characteristics of the relaxation of monotonic conditional swap functions, and propose several monotonic swap functions, e.g., the Cauchy and optimal monotonic functions (Petersen et al., 2022).

**Permutation-Equivariant Networks.** A seminal architecture, long short-term memory (Hochreiter & Schmidhuber, 1997) can be used in modeling a sequence without any difficulty, and a sequence-to-sequence model (Sutskever et al., 2014) can be employed to cope with a sequence. However, as discussed in the work by Vinyals et al. (2016), an unordered sequence can have good orderings, by analyzing the effects of permutation thoroughly. Zaheer et al. (2017) propose a permutation-invariant or permutation-equivariant network, named Deep Sets, and prove the permutation invariance and permutation equivariance of the proposed models. By utilizing the Transformer network (Vaswani et al., 2017), Lee et al. (2019) have proposed a permutation-equivariant network.

## 7 DISCUSSION

**Numerical Analysis on Our Methods and Their Hyperparameters.** We carry out numerical analyses on the effects of our methods, compared to a baseline method, i.e., Diffsort with the optimal monotonic sigmoid function. As reported in Table 4, we demonstrate that our methods better sorting performance compared to the baseline, which implies that our suggestions are effective in the sorting tasks. In these experiments, we follow the settings of the experiments described in Section 5.1. Moreover, we present numerical analyses on a balancing hyperparameter, steepness, and a learning rate in Sections K and L.

**Analysis on Performance Gains.** According to the results in Sections 5 and 7, we can argue that the error-free DSFs, our proposed loss, and the Transformer-based models contribute to better

Table 4: Study on comparisons of Diffsort with the optimal monotonic sigmoid function and our methods in the experiments on sorting the four-digit MNIST dataset.

| Method | Model | Sequence Length | | | | | |
|--------|-------|------|------|------|------|------|------|
| | | 3 | 5 | 7 | 9 | 15 | 32 |
| Diffsort | CNN | 94.6 (96.3) | 85.0 (93.3) | 73.6 (90.7) | 62.2 (88.5) | 31.8 (82.3) | 1.4 (67.9) |
| Ours | | 95.2 (96.7) | 87.2 (94.2) | 76.6 (91.6) | 64.8 (89.2) | 34.7 (83.3) | 2.1 (69.2) |
| Diffsort | Transformer-S | 95.9 (97.1) | 90.2 (95.4) | 83.9 (94.2) | 77.2 (92.9) | 57.3 (89.7) | 16.3 (81.7) |
| Ours | | 95.9 (97.1) | 94.8 (97.5) | 90.8 (96.5) | 86.9 (95.7) | 74.3 (93.6) | 37.8 (87.7) |
| Diffsort | Transformer-L | 96.5 (97.5) | 92.6 (96.4) | 87.6 (95.3) | 82.6 (94.3) | 67.8 (92.0) | 32.1 (85.7) |
| Ours | | 96.5 (97.5) | 95.4 (97.7) | 92.9 (97.2) | 90.1 (96.5) | 82.5 (95.0) | 46.2 (88.9) |

performance considerably compared to the baseline methods. As shown in Tables 1 and 4, the performance gains by the Transformer-based models are more substantial than the gains by the error-free DSFs and our loss, since multi-head attention is effective for capturing long-term dependency (or dependency between multiple instances in our case) and reducing inductive biases. However, as will be discussed in the following, the hard permutation matrices can be used in the case that does not allow us to mix instances in $\mathbf{X}$, e.g., sorting image fragments in Section 5.2.

**Utilization of Hard Permutation Matrices.** While the use of a soft permutation matrix $\mathbf{P}_{\text{soft}}$ makes given instances mixed, a hard permutation matrix $\mathbf{P}_{\text{hard}}$ is instrumental in applying $\mathbf{P}_{\text{hard}}$ in a problem that requires swapping given instances exactly. More precisely, each row of $\mathbf{P}_{\text{soft}}^{\top}\mathbf{X}$ is a linear combination of some column of $\mathbf{P}_{\text{soft}}$ and $\mathbf{X}$, but one of $\mathbf{P}_{\text{hard}}^{\top}\mathbf{X}$ corresponds to an exact row in $\mathbf{X}$. This property can be used to preserve input instances from sorting operations. The experiments in Section 5.2 can be considered as one of such cases, and it exhibits the strength of our method, not only the performance in $\text{acc}_{\text{em}}$ and $\text{acc}_{\text{ew}}$.

**Effects of Multi-Head Attention in the Problem** (3)**.** We follow the model architecture used in the previous work (Grover et al., 2019; Cuturi et al., 2019; Petersen et al., 2021; 2022) for the CNNs. However, as shown in Tables 1, 2, and 3, the model is not enough to show the best performance. In particular, whereas the model capacity, i.e., FLOPs and the number of parameters, of the Transformer-Small models is almost matched to or less than the capacity of the CNNs, the results by the Transformer-Small models outperform the results by the CNNs. We presume that these performance gains are derived from a multi-head attention's ability to capture long-term dependency and reduce inductive biases, as widely stated in many recent studies in diverse fields such as natural language processing (Vaswani et al., 2017; Devlin et al., 2018; Brown et al., 2020), computer vision (Dosovitskiy et al., 2021; Liu et al., 2021), and 3D vision (Nash et al., 2020; Zhao et al., 2021). Especially, unlike the instance-wise CNNs, our permutation-equivariant Transformer architecture utilizes self-attention for given instances, so that our model can productively compare instances in a sequence and effectively learn the relative relationship between them.

**Further Study of Differentiable Sorting Algorithms.** Differentiable sorting encourages us to train a mapping from an abstract input to an ordinal score using supervision on permutation matrices. However, this line of studies is limited to a sorting problem of high-dimensional data with clear ordering information, e.g., multi-digit numbers. As the further study of differentiable sorting, we can expand this framework to sort more ambiguous data, which contains implicitly ordinal information.

## 8 CONCLUSION

In this paper, we defined a softening error, induced by a monotonic DSF, and demonstrated several evidences of the error accumulation problem. To resolve the error accumulation problem, an error-free DSF is proposed, inspired by a straight-through estimator. Moreover, we provided the simple theoretical and empirical analyses that our error-free DSF successfully achieves a zero error and also holds a non-decreasing condition and differentiability. By combining all components, we suggested a generalized neural sorting network with the error-free DSF and multi-head attention. Finally, we showed that our methods are better than or comparable to other algorithms in diverse benchmarks.

ETHICS STATEMENT

As discussed in Section 7, the hard permutation matrices produced by our methods allow us to swap instances exactly, not the linear combination of instances. This characteristic is required when we are given the final outcomes of sorting as supervision. This scenario is tested by the experiments presented in Section 5.2. In these experiments, we are supposed that original images are provided as supervision. Building on the advantages of neural network-based sorting networks, we expand their practical significance to the cases that need hard permutation matrices. On the other hand, the nature of neural sorting networks may yield a potential negative societal impact. If this line of research including our approaches is employed to sort controversial high-dimensional data such as beauty and intelligence, it can be considered as the unethical use cases of artificial intelligence.

ACKNOWLEDGMENTS

This work was supported by the IITP grants (2022-0-00290: Visual Intelligence for Space-Time Understanding and Generation based on Multi-layered Visual Common Sense, 2022-0-00264: Comprehensive Video Understanding and Generation with Knowledge-based Deep Logic Neural Network) funded by Ministry of Science and ICT, Republic of Korea.

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

# A  OPTIMAL MONOTONIC SIGMOID FUNCTIONS

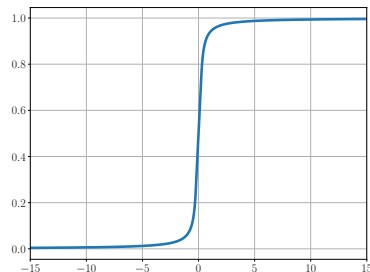

Figure 4: An optimal monotonic sigmoid function, which is presented in (7).

We visualize an optimal monotonic sigmoid function in Figure 4.

# B  COMPARISONS OF DIFFERENTIABLE SWAP FUNCTIONS

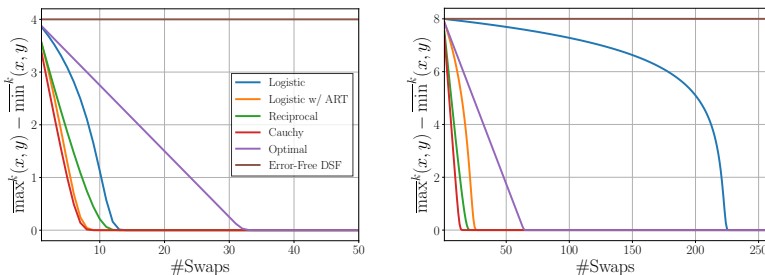

Figure 5: Comparisons of diverse DSFs in terms of the numbers of swap functions applied. Our error-free DSF does not change the original $x$ and $y$, unlike other DSFs. We initially set $x = 4, y = 0$ for the left panel or $x = 8, y = 0$ for the right panel, where $k = \#\text{Swaps}$.

As depicted in Figure 5, some sigmoid functions such as the logistic, logistic with ART, reciprocal, Cauchy, and optimal monotonic functions suffer from the error accumulation problem; see the work (Petersen et al., 2022) for the details of such sigmoid functions. For the case of the Cauchy function, two values are close enough at the 9th step in the left panel of Figure 5 and the 15th step in the right panel of Figure 5; we calculate the corresponding steps where a difference between two values becomes smaller than 0.001.

# C  COMPARISONS OF DIFFERENT SORTING NETWORKS

Figure 6 shows the comparisons of different sorting networks by varying sequence lengths. $\text{acc}_{\text{em}}$ and $\text{acc}_{\text{ew}}$ are measured to assess the sorting networks.

# D  PROOF OF PROPOSITION 1

*Proof.* If two elements at indices $i$ and $j$ are swapped by a single swap function, $[\mathbf{P}]_{kk} = 1$ for $k \in [n] \backslash \{i, j\}$, $[\mathbf{P}]_{kl} = 0$ for $k \neq l, k, l \in [n] \backslash \{i, j\}$, $[\mathbf{P}]_{ii} = [\mathbf{P}]_{jj} = p$, and $[\mathbf{P}]_{ij} = [\mathbf{P}]_{ji} = 1 - p$, where $p$ is the output of a sigmoid function, i.e., $p = \sigma(y - x)$ or $p = \lfloor \sigma(y - x) \rceil$. Since the multiplication of doubly-stochastic matrices is still doubly-stochastic, Proposition 1 is true. $\qquad\square$

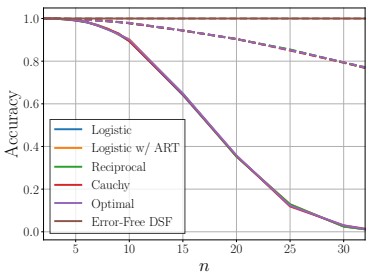

Figure 6: $\mathrm{acc_{em}}$ (solid) and $\mathrm{acc_{ew}}$ (dashed), versus sequence lengths. Without a mapping $g$ from $\mathbf{x}$ to $s$, we uniformly sample $n$ elements from $[-10, 10]$, which are considered as ordinal scores $\mathbf{s}$. Then, we sort them using the respective sorting networks and measure accuracy over the ground-truth permutation matrices computed by $n$ elements. We repeat each experiment 10,000 times.

## E   PROOF OF PROPOSITION 2

*Proof.* Let $\overline{\min}^k(x, y)$ and $\overline{\min}^k(x, y)$ be minimum and maximum values where swap with $\overline{\min}$ and $\overline{\max}$ is applied $k$ times repeatedly. By Definition 1 and $\overline{\min}^i < \overline{\min}^{i+1}$ and $\overline{\max}^{i+1} < \overline{\max}^i$, the following inequalities are satisfied:

$$\min(x, y) < \overline{\min}^1(x, y) < \overline{\min}^2(x, y) < \cdots < \overline{\min}^k(x, y)$$
$$\leq \overline{\max}^k(x, y) < \cdots < \overline{\max}^2(x, y) < \overline{\max}^1(x, y) < \max(x, y), \tag{21}$$

under the assumption that $\nabla_x \sigma(x) > 0$. By (21) and $\nabla_x \sigma(x) > 0$, we can obtain the following inequality:

$$0 \leq \overline{\max}^{k+1}(x, y) - \overline{\min}^{k+1}(x, y) < \overline{\max}^k(x, y) - \overline{\min}^k(x, y) < \overline{\max}^{k-1}(x, y) - \overline{\min}^{k-1}(x, y). \tag{22}$$

Therefore, $\lim_{k \to \infty} \overline{\max}^k(x, y) - \overline{\min}^k(x, y) = 0$, and $\overline{\min}^k(x, y) = \overline{\max}^k(x, y)$ if $k \to \infty$. To sum up, a softening error for $k \to \infty$ is $(\max(x, y) - \min(x, y))/2$ since $\overline{\max}^k(x, y) = (\min(x, y) + \max(x, y))/2$ by (9). Note that the assumption $\nabla_x \sigma(x) > 0$ implies that $\sigma(\cdot)$ is a strictly monotonic sigmoid function. □

## F   PROOF OF PROPOSITION 3

*Proof.* According to Definition 1, given $x$ and $y$, the softening error $x' - \min(x, y)$ is expressed as the following:

$$x' - \min(x, y) = \left(\min(x, y) - \overline{\min}(x, y)\right)_{\mathrm{sg}} + \overline{\min}(x, y) - \min(x, y)$$
$$= \min(x, y) - \overline{\min}(x, y) + \overline{\min}(x, y) - \min(x, y)$$
$$= 0, \tag{23}$$

while a forward pass is applied. The proof for $\max(x, y) - y'$ is omitted because it is obvious. □

## G   DETAILS OF PERMUTATION-EQUIVARIANT NETWORKS WITH MULTI-HEAD ATTENTION

Figure 7 illustrates the Transformer-based permutation-equivariant network, which is implemented with multi-head attention (Vaswani et al., 2017). For the sake of brevity, this illustration briefly depicts our permutation-equivariant network without detailing the specifics of the Transformer network. Each instance in a sequence is first processed by a feature extractor, i.e., a convolutional neural network. Then, a sequence of latent vectors is provided into the Transformer network without positional encoding. At each multi-head attention module, each latent vector is updated by the aggregation of the latent vectors given where the aggregation is determined by the operations explained in Section 4 and (16). After passing through multiple layers of multi-head attention and the

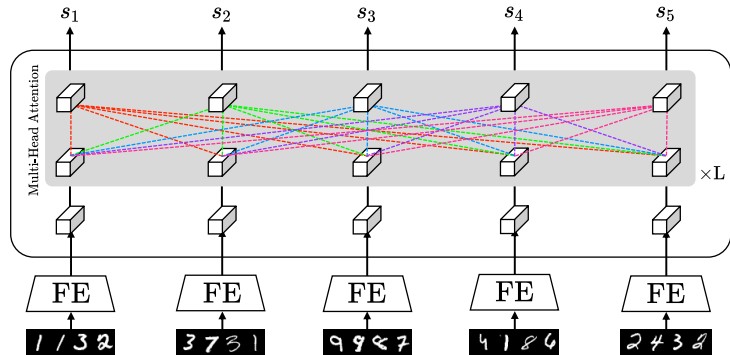

Figure 7: Simple illustration of our permutation-equivariant network with multi-head attention. For the sake of brevity, we do not depict the details of multi-head attention, layer normalization, and feed-forward networks. Note that FE stands for a feature extractor.

corresponding components such as layer normalization and feed-forward neural networks, the final fully-connected layer is applied to transform the outputs of the Transformer network into a score vector $\mathbf{s}$. The details of the Transformer network can be found in the work by Vaswani et al. (2017).

## H  SPLIT STRATEGY TO REDUCE THE NUMBER OF POSSIBLE PERMUTATIONS

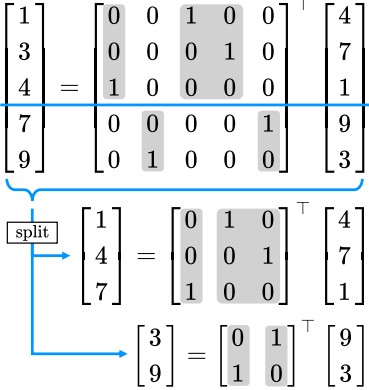

Figure 8: A split process of $\mathbf{P}$.

As presented in (14) and Proposition 1, the permutation matrix for the error-free DSF is a discretized doubly-stochastic matrix, which is denoted as $\mathbf{P}_{\mathrm{hard}}$, in a forward pass, and is differentiable in a backward pass. Here, we show an interesting proposition of $\mathbf{P}_{\mathrm{hard}}$:

**Proposition 4.** *Let* $\mathbf{s} \in \mathbb{R}^n$ *and* $\mathbf{P}_{\mathrm{hard}} \in \mathbb{R}^{n \times n}$ *be an unordered sequence and the corresponding permutation matrix to transform it to* $\mathbf{s}_\mathrm{o}$, *respectively. We are able to split* $\mathbf{s}$ *to two subsequences* $\mathbf{s}_1 \in \mathbb{R}^{n_1}$ *and* $\mathbf{s}_2 \in \mathbb{R}^{n_2}$ *where* $\mathbf{s}_1 = [\mathbf{s}]_{1:n_1}$ *and* $\mathbf{s}_2 = [\mathbf{s}]_{n_1+1:n_1+n_2}$. *Then,* $\mathbf{P}_{\mathrm{hard}}$ *is also split to* $\mathbf{P}_1 \in \mathbb{R}^{n_1 \times n_1}$ *and* $\mathbf{P}_2 \in \mathbb{R}^{n_2 \times n_2}$, *so that* $\mathbf{P}_1$ *and* $\mathbf{P}_2$ *are (discretized) doubly-stochastic.*

*Proof.* A split does not change the relative order of elements in the same split and each entry in the permutation matrix is zero or one, so that a permutation matrix can be split as shown in Figure 8. Moreover, multiple splits are straightforwardly doable. □

In contrast to $\mathbf{P}_{\mathrm{hard}}$, it is impossible to split $\mathbf{P}_{\mathrm{soft}}$ to sub-block matrices since such sub-block matrices cannot satisfy the property of doubly-stochastic matrix, which is discussed in Proposition 1. Importantly, Proposition 4 does not show a possibility of the recoverable decomposition of the permutation matrix, which implies that we cannot guarantee the recovery of decomposed matrices to

the original matrix. Regardless of the existence of recoverable decomposition, we attempt to reduce the number of possible permutations with sub-block matrices, rather than holding the large number of possible permutations with the original permutation matrix. Therefore, by Proposition 4, relative relationships between instances with a smaller number of possible permutations are more distinctively learnable than the relationships with a larger number of possible permutations, preventing a sparse correct permutation among a large number of possible permutations.

## I    DETAILS OF ARCHITECTURES

We describe the details of the neural architectures used in our paper, as shown in Tables 5, 6, 7, 8, 9, 10, 11, 12, 13, and 14. For the experiments on sorting image fragments, we omit some of the architectures employed for particular fragmentation, since they follow the same architectures presented in Tables 11, 12, 13, and 14. Only differences are the sizes of inputs, and therefore the respective sizes of the first fully-connected layers change.

Table 5: Architecture of the convolutional neural networks for the four-digit MNIST dataset.

| Layer | Input & Output (Channel) Dimensions | Kernel Size | Details |
|---|---|---|---|
| Convolutional | $1 \times 32$ | $5 \times 5$ | strides 1, padding 2 |
| ReLU | – | – | – |
| Max-pooling | – | – | pooling 2, strides 2 |
| Convolutional | $32 \times 64$ | $5 \times 5$ | strides 1, padding 2 |
| ReLU | – | – | – |
| Max-pooling | – | – | pooling 2, strides 2 |
| Fully-connected | $12544 \times 64$ | – | – |
| ReLU | – | – | – |
| Fully-connected | $64 \times 1$ | – | – |

Table 6: Architecture of the Transformer-Small models for the four-digit MNIST dataset.

| Layer | Input & Output (Channel) Dimensions | Kernel Size | Details |
|---|---|---|---|
| Convolutional | $1 \times 32$ | $5 \times 5$ | strides 1, padding 2 |
| ReLU | – | – | – |
| Max-pooling | – | – | pooling 2, strides 2 |
| Convolutional | $32 \times 64$ | $5 \times 5$ | strides 1, padding 2 |
| ReLU | – | – | – |
| Max-pooling | – | – | pooling 2, strides 2 |
| Fully-connected | $12544 \times 16$ | – | – |
| Transformer Encoder | $16 \times 16$ | | #layers 6, #heads 8 |
| ReLU | – | – | – |
| Fully-connected | $16 \times 1$ | – | – |

Table 7: Architecture of the Transformer-Large models for the four-digit MNIST dataset.

| Layer | Input & Output (Channel) Dimensions | Kernel Size | Details |
|---|---|---|---|
| Convolutional | $1 \times 32$ | $5 \times 5$ | strides 1, padding 2 |
| ReLU | – | – | – |
| Max-pooling | – | – | pooling 2, strides 2 |
| Convolutional | $32 \times 64$ | $5 \times 5$ | strides 1, padding 2 |
| ReLU | – | – | – |
| Max-pooling | – | – | pooling 2, strides 2 |
| Fully-connected | $12544 \times 64$ | – | – |
| Transformer Encoder | $64 \times 64$ | | #layers 8, #heads 8 |
| ReLU | – | – | – |
| Fully-connected | $64 \times 1$ | – | – |

## J    DETAILS OF EXPERIMENTS

As described in the main article, we use three public datasets: MNIST (LeCun et al., 1998), SVHN (Netzer et al., 2011), and CIFAR-10 (Krizhevsky & Hinton, 2009). Unless otherwise spec-

ified, a learning rate $10^{-3.5}$ is used for the CNN architectures and a learning rate $10^{-4}$ is used for the Transformer-based architectures; see our implementation for the exact learning rates we utilize in the experiments. Learning rate decay is applied by multiplying $0.5$ in every 50,000 steps for the experiments on sorting multi-digit images and every 20,000 steps for the experiments on sorting image fragments. Moreover, we balance two objectives for $\mathbf{P}_{\text{hard}}$ and $\mathbf{P}_{\text{soft}}$ by multiplying $1, 0.1$, $0.01$, or $0.001$; see our implementation for the respective values for all the experiments. For random seeds, we pick five random seeds $42, 84, 126, 168$, and $210$ for all the experiments; these values are picked without any trials. Other missing details can be found in our implementation. Furthermore, we employ several commercial NVIDIA GPUs, i.e., GeForce GTX Titan Xp, GeForce RTX 2080, and GeForce RTX 3090, in the experiments.

## K  STUDY ON BALANCING HYPERPARAMETER

We conduct a study on a balancing hyperparameter in the experiments on sorting the four-digit MNIST dataset, as shown in Table 15. For these experiments, we use steepness $2, 14, 23, 38, 25$, and $124$ for sequence lengths $3, 5, 7, 9, 15$, and $32$, respectively. Also, we use a learning rate $10^{-3}$ and 5 random seeds $42, 84, 126, 168$, and $210$.

## L  STUDY ON STEEPNESS AND LEARNING RATE

We present studies on steepness and learning rate for the experiments on sorting the multi-digit MNIST dataset, as shown in Tables 16, 17, 18, 19, 20, and 21. For these experiments, a random seed 42 is only used due to numerous experimental settings. Also, we use balancing hyperparameters $\lambda$ as $1.0, 1.0, 0.1, 0.1, 0.1$, and $0.1$ for sequence lengths $3, 5, 7, 9, 15$, and $32$, respectively. Since there are many configurations of steepness, learning rate, and a balancing hyperparameter, we cannot include all the configurations here. The final configurations we use in the experiments are described in our implementation. As widely known in the deep learning community, a learning rate should be set as a value around $10^{-3}$. Moreover, according to our empirical analyses, steepness should generally be higher as a sequence length is longer.

## M  LIMITATIONS

While a sorting task is one of the most significant problems in computer science and mathematics (Cormen et al., 2022), our ideas, which are built on sorting networks (Knuth, 1998; Ajtai et al., 1983), can be limited to sorting algorithms. It implies that it is not easy to devise neural network-based approaches to solving general problems in computer science, e.g., combinatorial optimization, which are inspired by our ideas.

In addition, while our proposed methods show the superior performance compared to the baseline methods, this line of research suffers from performance degradation for longer sequences as shown in Tables 1, 2, and 3. More precisely, for longer sequences, the element-wise accuracy does not decline dramatically, but the sequence-wise accuracy significantly drops due to the nature of sequences. Incorporating our contributions such as the error-free DSFs and the Transformer-based networks, we expect that the further progress of neural network-based sorting networks can be achieved. In particular, the consideration of more sophisticated neural networks, which are capable of handling longer sequences, might help improve performance. This will be left for future work.

Our frameworks successfully learn relationships between high-dimensional data with ordinal contents as shown in Section 5. However, we suppose that our methods might fail in sorting data without ordinal information; the elaborate discussion on this topic can be found in Section 7. In order to sort more ambiguous high-dimensional data, we can combine our work with part-based or segmentation-based approaches.

Table 8: Architecture of the convolutional neural networks for the SVHN dataset.

| Layer | Input & Output (Channel) Dimensions | Kernel Size | Details |
|---|---|---|---|
| Convolutional | $3 \times 32$ | $5 \times 5$ | strides 1, padding 2 |
| ReLU | – | – | – |
| Max-pooling | – | – | pooling 2, strides 2 |
| Convolutional | $32 \times 64$ | $5 \times 5$ | strides 1, padding 2 |
| ReLU | – | – | – |
| Max-pooling | – | – | pooling 2, strides 2 |
| Convolutional | $64 \times 128$ | $5 \times 5$ | strides 1, padding 2 |
| ReLU | – | – | – |
| Max-pooling | – | – | pooling 2, strides 2 |
| Convolutional | $128 \times 256$ | $5 \times 5$ | strides 1, padding 2 |
| ReLU | – | – | – |
| Max-pooling | – | – | pooling 2, strides 2 |
| Fully-connected | $2304 \times 64$ | – | – |
| ReLU | – | – | – |
| Fully-connected | $64 \times 1$ | – | – |

Table 9: Architecture of the Transformer-Small models for the SVHN dataset.

| Layer | Input & Output (Channel) Dimensions | Kernel Size | Details |
|---|---|---|---|
| Convolutional | $3 \times 32$ | $5 \times 5$ | strides 1, padding 2 |
| ReLU | – | – | – |
| Max-pooling | – | – | pooling 2, strides 2 |
| Convolutional | $32 \times 64$ | $5 \times 5$ | strides 1, padding 2 |
| ReLU | – | – | – |
| Max-pooling | – | – | pooling 2, strides 2 |
| Convolutional | $64 \times 64$ | $5 \times 5$ | strides 1, padding 2 |
| ReLU | – | – | – |
| Max-pooling | – | – | pooling 2, strides 2 |
| Convolutional | $64 \times 128$ | $5 \times 5$ | strides 1, padding 2 |
| ReLU | – | – | – |
| Max-pooling | – | – | pooling 2, strides 2 |
| Fully-connected | $1152 \times 32$ | – | – |
| Transformer Encoder | $32 \times 32$ | | #layers 6, #heads 8 |
| ReLU | – | – | – |
| Fully-connected | $32 \times 1$ | – | – |

Table 10: Architecture of the Transformer-Large models for the SVHN dataset.

| Layer | Input & Output (Channel) Dimensions | Kernel Size | Details |
|---|---|---|---|
| Convolutional | $3 \times 32$ | $5 \times 5$ | strides 1, padding 2 |
| ReLU | – | – | – |
| Max-pooling | – | – | pooling 2, strides 2 |
| Convolutional | $32 \times 64$ | $5 \times 5$ | strides 1, padding 2 |
| ReLU | – | – | – |
| Max-pooling | – | – | pooling 2, strides 2 |
| Convolutional | $64 \times 128$ | $5 \times 5$ | strides 1, padding 2 |
| ReLU | – | – | – |
| Max-pooling | – | – | pooling 2, strides 2 |
| Convolutional | $128 \times 256$ | $5 \times 5$ | strides 1, padding 2 |
| ReLU | – | – | – |
| Max-pooling | – | – | pooling 2, strides 2 |
| Fully-connected | $2304 \times 64$ | – | – |
| Transformer Encoder | $64 \times 64$ | | #layers 8, #heads 8 |
| ReLU | – | – | – |
| Fully-connected | $64 \times 1$ | – | – |

Table 11: Architecture of the convolutional neural networks for the MNIST dataset of 4 image fragments of size $14 \times 14$.

| Layer | Input & Output (Channel) Dimensions | Kernel Size | Details |
|---|---|---|---|
| Convolutional | $1 \times 32$ | $3 \times 3$ | strides 2, padding 1 |
| ReLU | – | – | – |
| Convolutional | $32 \times 64$ | $3 \times 3$ | strides 2, padding 1 |
| ReLU | – | – | – |
| Fully-connected | $1024 \times 64$ | – | – |
| ReLU | – | – | – |
| Fully-connected | $64 \times 1$ | – | – |

Table 12: Architecture of the Transformer models for the MNIST dataset of 4 image fragments of size $14 \times 14$.

| Layer | Input & Output (Channel) Dimensions | Kernel Size | Details |
|---|---|---|---|
| Convolutional | $1 \times 32$ | $3 \times 3$ | strides 2, padding 1 |
| ReLU | – | – | – |
| Convolutional | $32 \times 32$ | $3 \times 3$ | strides 2, padding 1 |
| ReLU | – | – | – |
| Fully-connected | $512 \times 16$ | – | – |
| Transformer Encoder | $16 \times 16$ | | #layers 1, #heads 8 |
| ReLU | – | – | – |
| Fully-connected | $16 \times 1$ | – | – |

Table 13: Architecture of the convolutional neural networks for the CIFAR-10 dataset of 4 image fragments of size $16 \times 16$.

| Layer | Input & Output (Channel) Dimensions | Kernel Size | Details |
|---|---|---|---|
| Convolutional | $3 \times 32$ | $3 \times 3$ | strides 2, padding 1 |
| ReLU | – | – | – |
| Convolutional | $32 \times 64$ | $3 \times 3$ | strides 2, padding 1 |
| ReLU | – | – | – |
| Fully-connected | $1024 \times 64$ | – | – |
| ReLU | – | – | – |
| Fully-connected | $64 \times 1$ | – | – |

Table 14: Architecture of the Transformer models for the CIFAR-10 dataset of 4 image fragments of size $16 \times 16$.

| Layer | Input & Output (Channel) Dimensions | Kernel Size | Details |
|---|---|---|---|
| Convolutional | $3 \times 32$ | $3 \times 3$ | strides 2, padding 1 |
| ReLU | – | – | – |
| Convolutional | $32 \times 32$ | $3 \times 3$ | strides 2, padding 1 |
| ReLU | – | – | – |
| Fully-connected | $512 \times 16$ | – | – |
| Transformer Encoder | $16 \times 16$ | | #layers 1, #heads 8 |
| ReLU | – | – | – |
| Fully-connected | $16 \times 1$ | – | – |

Table 15: Study on a balancing hyperparameter $\lambda$ in the experiments on sorting the four-digit MNIST dataset.

| $\lambda$ | Sequence Length | | | | | |
|---|---|---|---|---|---|---|
| | 3 | 5 | 7 | 9 | 15 | 32 |
| 1.000 | 94.8 (96.4) | 86.9 (94.1) | 74.2 (90.9) | 62.6 (88.6) | 12.0 (69.4) | 0.0 (38.9) |
| 0.100 | 94.9 (96.5) | 87.2 (94.2) | 75.3 (91.3) | 64.7 (89.2) | 34.7 (83.3) | 2.1 (69.2) |
| 0.010 | 95.1 (96.6) | 87.1 (92.7) | 75.2 (91.2) | 63.5 (88.8) | 33.2 (82.7) | 0.7 (60.7) |
| 0.001 | 95.2 (96.7) | 87.0 (94.1) | 76.6 (91.6) | 64.8 (89.2) | 32.9 (82.7) | 0.5 (61.5) |
| 0.000 | 94.9 (96.5) | 87.2 (94.2) | 75.9 (91.5) | 64.4 (89.1) | 34.1 (83.2) | 0.9 (60.6) |

Table 16: Study on steepness and learning rate for a sequence length 3. lr stands for learning rate.

| log₁₀ lr | Steepness | | | | | | |
|---|---|---|---|---|---|---|---|
| | 2 | 4 | 6 | 8 | 10 | 12 | 14 |
| -4.0 | 89.2 (92.6) | 91.5 (94.2) | 92.1 (94.6) | 92.3 (94.7) | 92.7 (95.0) | 92.1 (94.6) | 92.7 (95.0) |
| -3.5 | 93.6 (95.5) | 93.4 (95.5) | 94.5 (96.2) | 93.9 (95.9) | 94.4 (96.1) | 94.5 (96.2) | 94.6 (96.3) |
| -3.0 | 95.3 (96.8) | 95.1 (96.6) | 95.0 (96.5) | 94.4 (96.2) | 94.3 (96.1) | 94.7 (96.4) | 94.8 (96.5) |
| -2.5 | 94.5 (96.2) | 94.3 (96.1) | 93.7 (95.6) | 93.8 (95.7) | 93.7 (95.7) | 93.7 (95.6) | 94.1 (95.9) |

Table 17: Study on steepness and learning rate for a sequence length 5. lr stands for learning rate.

| log₁₀ lr | Steepness | | | | | | |
|---|---|---|---|---|---|---|---|
| | 14 | 16 | 18 | 20 | 22 | 24 | 26 |
| -4.0 | 78.3 (90.2) | 76.3 (89.3) | 79.5 (90.9) | 78.7 (90.4) | 78.4 (90.4) | 77.3 (89.8) | 79.0 (90.6) |
| -3.5 | 85.1 (93.3) | 83.6 (92.7) | 84.8 (93.1) | 85.5 (93.5) | 83.4 (92.5) | 85.6 (93.6) | 84.1 (92.8) |
| -3.0 | 86.9 (94.1) | 85.2 (93.3) | 86.2 (93.8) | 85.7 (93.6) | 85.4 (93.5) | 85.0 (93.2) | 85.0 (93.3) |
| -2.5 | 83.1 (92.3) | 83.2 (92.4) | 83.1 (92.4) | 81.7 (91.8) | 83.3 (92.5) | 82.4 (92.1) | 82.5 (92.1) |

Table 18: Study on steepness and learning rate for a sequence length 7. lr stands for learning rate.

| log₁₀ lr | Steepness | | | | | | |
|---|---|---|---|---|---|---|---|
| | 23 | 25 | 27 | 29 | 31 | 33 | 35 |
| -4.0 | 53.5 (82.9) | 57.3 (84.6) | 61.0 (86.0) | 58.3 (85.1) | 66.1 (88.0) | 57.3 (84.6) | 61.6 (86.4) |
| -3.5 | 71.6 (90.0) | 73.5 (90.8) | 68.1 (88.6) | 72.9 (90.5) | 72.2 (90.2) | 71.4 (89.9) | 74.2 (91.0) |
| -3.0 | 74.4 (90.9) | 72.8 (90.4) | 74.2 (90.8) | 69.2 (89.0) | 69.9 (89.3) | 73.0 (90.4) | 73.0 (90.5) |
| -2.5 | 68.0 (88.6) | 67.2 (88.1) | 68.1 (88.6) | 64.5 (86.9) | 66.8 (88.0) | 70.4 (89.3) | 66.2 (87.9) |

Table 19: Study on steepness and learning rate for a sequence length 9. lr stands for learning rate.

| log₁₀ lr | Steepness | | | | | | |
|---|---|---|---|---|---|---|---|
| | 26 | 28 | 30 | 32 | 34 | 36 | 38 |
| -4.0 | 34.0 (77.7) | 37.8 (79.7) | 37.3 (79.4) | 45.9 (83.0) | 46.4 (83.1) | 36.2 (78.9) | 45.3 (82.5) |
| -3.5 | 51.8 (84.8) | 59.7 (87.5) | 58.3 (87.5) | 61.0 (88.2) | 60.2 (88.1) | 54.6 (85.8) | 56.3 (86.6) |
| -3.0 | 62.5 (88.8) | 60.5 (88.0) | 61.8 (88.4) | 63.2 (88.9) | 61.5 (88.2) | 62.2 (88.6) | 65.0 (89.5) |
| -2.5 | 57.4 (86.8) | 58.7 (87.2) | 56.5 (86.4) | 55.3 (86.1) | 52.8 (85.0) | 46.9 (82.1) | 52.1 (84.6) |

Table 20: Study on steepness and learning rate for a sequence length 15. lr stands for learning rate.

| log₁₀ lr | Steepness | | | | | | |
|---|---|---|---|---|---|---|---|
| | 19 | 21 | 23 | 25 | 27 | 29 | 31 |
| -4.0 | 3.6 (62.7) | 7.2 (67.6) | 4.9 (64.8) | 2.6 (60.9) | 2.9 (61.5) | 2.5 (60.2) | 6.8 (68.3) |
| -3.5 | 10.3 (71.5) | 11.9 (73.0) | 7.6 (68.6) | 22.3 (78.7) | 7.1 (68.7) | 8.2 (69.6) | 10.6 (71.8) |
| -3.0 | 24.2 (79.8) | 24.1 (80.1) | 29.7 (81.9) | 32.1 (82.7) | 23.9 (79.9) | 31.6 (82.1) | 31.0 (82.0) |
| -2.5 | 30.5 (81.4) | 28.2 (80.0) | 18.9 (76.8) | 29.8 (80.9) | 19.8 (77.8) | 15.4 (75.4) | 24.6 (79.6) |

Table 21: Study on steepness and learning rate for a sequence length 32. lr stands for learning rate.

| log₁₀ lr | Steepness | | | | | | |
|---|---|---|---|---|---|---|---|
| | 118 | 120 | 122 | 124 | 126 | 128 | 130 |
| -4.0 | 0.0 (46.4) | 0.0 (45.8) | 0.0 (46.4) | 0.0 (43.1) | 0.0 (42.9) | 0.0 (49.4) | 0.0 (48.9) |
| -3.5 | 0.2 (60.5) | 0.3 (61.8) | 0.7 (64.8) | 0.3 (62.7) | 0.0 (56.0) | 0.3 (62.7) | 0.2 (59.8) |
| -3.0 | 0.6 (63.1) | 0.5 (63.6) | 0.4 (59.8) | 0.8 (65.8) | 0.2 (58.1) | 0.1 (56.0) | 0.1 (57.1) |
| -2.5 | 0.5 (62.8) | 0.5 (62.3) | 0.5 (62.5) | 0.1 (58.3) | 0.3 (61.2) | 0.1 (59.5) | 0.2 (58.1) |

## N    ADDITIONAL DISCUSSION

It is challenging to directly sort a sequence of generic data instances without using auxiliary networks and explicit supervision. Unlike earlier sorting methods, this sorting network-based research (Petersen et al., 2021; 2022) including our work ensures that we can train a neural network that predicts numerical scores and eventually sorts them, even though we do not necessitate accessing explicit supervision such as exact numerical values of the contents in high-dimensional data. In this sense, the practical significance of our proposed methods can be highlighted by offering this possibility of solving a sorting problem with high-dimensional inputs. For example, as shown in Section 5, we can compare images of street view house numbers using the sorting network where our neural network is trained without exact house numbers.

Moreover, instead of using costly supervision, our networks allow us to sort high-dimensional instances in a sequence where information on comparisons between instances is only given. This scenario often occurs when we cannot obtain complete supervision. For example, if we would sort four-digit MNIST images, ordinary neural networks are designed to solve a classification task by predicting class probabilities each of which indicates one of all labels from "0000" to "9999". If some labels are missing and further we do not know the exact number of labels, they might fail in predicting unseen data corresponding to those labels. Unlike these methods, it is possible to solve sorting problems using our networks in such a scenario.

Furthermore, this study can be applied in diverse deep learning tasks for learning to sort generic high-dimensional data, such as information retrieval (Cao et al., 2007; Liu, 2009) and top-$k$ classification (Berrada et al., 2018).

