# OpenReview forum: "Generalized Neural Sorting Networks with Error-Free Differentiable Swap Functions"
_ICLR.cc/2024/Conference — ICLR 2024 poster_

### Official Review · Reviewer_M9Vx · 2023-10-29

**Soundness:** 3 good
**Presentation:** 3 good
**Contribution:** 3 good
**Rating:** 6
**Confidence:** 3

**Summary:**

The authors investigate the properties of differentiable swap functions for neural sorting. They conclude that the existing methods shrink the difference between different elements at each application of the swap function. This has a negative effect on the performance of such networks. To alleviate this problem, the authors propose an Error-Free Differentiable Swap Function, based on the straight-through estimator. Also, they propose to use transformers without positional encodings for processing the raw inputs before they are fed to

**Strengths:**

- Simple method
- Better performance on long sequences
- Motivated by analyzing existing methods

**Weaknesses:**

- No clear distinction between the Transformer-based network vs the differentiable swap function. These are two orthogonal changes, and the Transformer seems to have much more performance impact than the swap function itself. Yet, the paper focuses mostly on the swap function.
- The method uses both the soft and the hard sorting network for training. Also, Tab 15 shows that training with $\lambda = 0$ has basically no effect on the quality of the trained network. As the standard deviations are not shown, it is unclear whether there is any effect here, but $\lambda=1$ seems to hinder accuracy. This begs the question if the Error-Free Differentiable Swap Function has any contribution to training, or just during inference, in which case there is no need for the straight-through estimator. It would be worth evaluating the baseline Optimal Diffsort with hard min/max during inference, to see this.

Based on the 2 points above, I'd like to see two sets of experiments:
- the best optimal diff sort models with identical transformer architecture to those of the Error-Free DSF
- The same architecture as the best Error-Free DSF with \lambda=0 (thus, being equivalent to using hard min/max during inference, but being trained as "Diffsort Optimal")

**Questions:**

Why are the results on 4-digit MNIST with a seq length of 32 significantly better in Tab 1 than in Tab 15?

---

> ### Author Response · Authors · 2023-11-15
> **Response to Reviewer M9Vx (1/n)**
>
> We appreciate your constructive comment to improve our work.
>
> > No clear distinction between the Transformer-based network vs the differentiable swap function. These are two orthogonal changes, and the Transformer seems to have much more performance impact than the swap function itself. Yet, the paper focuses mostly on the swap function.
>
> We agree with your point. To accommodate your comment, we have updated **Section 4** and added **Section G** and **Figure 7**. These updates explain our Transformer-based network in detail. Please see our revision as well as the discussion described in the **Effects of Multi-Head Attention in the Problem (3)** paragraph of **Section 7**.
>
> >  As the standard deviations are not shown,
>
> We provide the standard deviations here. These results show the standard deviations of the last three rows of **Table 1**.
>
> |   | 3 | 5 | 7 | 9 | 15 | 32 |
> |---|---|---|---|---|----|----|
> | CNN | 0.41 (0.29) | 1.08 (0.44) | 2.12 (0.76) | 2.74 (0.85) | 2.75 (0.81) | 0.40 (0.49) |
> | Transformer-S | 0.20 (0.16) | 0.21 (0.11) | 0.57 (0.22) | 0.56 (0.24) | 0.99 (0.30) | 1.54 (0.46) |
> | Transformer-L | 0.27 (0.18) | 0.35 (0.17) | 0.52 (0.24) | 0.36 (0.18) | 0.76 (0.20) | 1.14 (0.34) |
>
> Based on the values shown above, the training of our networks is dependent on random seeds but the dependency is not strong. To consistently present the experimental results, we omitted them in the submission.
>
> > $\lambda = 1$ seems to hinder accuracy.
>
> You are right, but the use of our hard loss is effective for $0 < \lambda < 1$. As described in **Section 7**, our hard loss is likely to make loss values discrete. Therefore, the selection of appropriate balancing parameters is significant. We would like to note that this selection process is a typical problem in the deep learning community,
> and we also solved a similar problem by following the previous literature.
>
> > the best optimal diffsort models with identical transformer architecture to those of the Error-Free DSF
>
> If we understand your concern correctly, we have already included this ablation study. We can see these results in **Table 4**. For your convenience, we highlight these results here.
>
> | Method | Model | 3 | 5 | 7 | 9 | 15 | 32 |
> |--------|-------|---|---|---|---|----|----|
> | Diffsort | CNN | 94.6 (96.3) | 85.0 (93.3) | 73.6 (90.7) | 62.2 (88.5) | 31.8 (82.3) | 1.4 (67.9) |
> | Ours | CNN | 94.8 (96.4) | 86.9 (94.1) | 74.2 (90.9) | 62.6 (88.6) | 34.7 (83.3) | 2.1 (69.2) |
> | Diffsort | Transformer-S | 95.9 (97.1) | 90.2 (95.4) | 83.9 (94.2) | 77.2 (92.9) | 57.3 (89.7) | 16.3 (81.7) |
> | Ours | Transformer-S | 95.9 (97.1) | 94.8 (97.5) | 90.8 (96.5) | 86.9 (95.7) | 74.3 (93.6) | 37.8 (87.7) |
> | Diffsort | Transformer-L | 96.5 (97.5) | 92.6 (96.4) | 87.6 (95.3) | 82.6 (94.3) | 67.8 (92.0) | 32.1 (85.7) |
> | Ours | Transformer-L | 96.5 (97.5) | 95.4 (97.7) | 92.9 (97.2) | 90.1 (96.5) | 82.5 (95.0) | 46.2 (88.9) |
>
> > The same architecture as the best Error-Free DSF with $\lambda = 0$ (thus, being equivalent to using hard min/max during inference, but being trained as "Diffsort Optimal")
>
> Since we directly calculated accuracy from the scores $\mathbf{s}$ produced by the neural network (see **Equations (19) and (20)**), the results shown in the paper (including Diffsort (Optimal)) are reported by following your suggestion, which uses hard min/max during inference.
>
> > Why are the results on 4-digit MNIST with a seq length of 32 significantly better in Tab 1 than in Tab 15?
>
> **Table 15** uses fixed hyperparameters (i.e., steepness and learning rate) for fair comparison, so that it can show a thorough analysis on our models. Therefore, the results in **Table 1** is better than ones in **Table 15**. Please see **Section K** for more details.

---

> > ### Comment · Reviewer_M9Vx · 2023-11-16
> >
> > Thank you for the clarifications. This clears most of my doubts. My remaining concern is about Tab 15. Is there any reason why it does not use better hyperparameters from Table 1 as the fixed hyperparameters? Reporting ablations on a weaker hyperparameter set casts doubts on the validity of the trends that we can conclude from the results.

---

> > > ### Author Response · Authors · 2023-11-16
> > >
> > > We are pleased that most of your doubts are resolved.
> > >
> > > The hyperparameters used in Table 15 came from the previous work (Petersen et al., 2022). When we designed this study, we intended to compare our results to the previous work. Thus, we used those hyperparameters. If you think that a new study with better hyperparameters can improve the understanding of this study, we will re-do the experiments.

---

> > > > ### Comment · Reviewer_M9Vx · 2023-11-16
> > > >
> > > > Please include the experiments with the new hyperparameters in the final version of the paper.
> > > >
> > > > Given that most of my doubts are resolved, I'm increasing my score.

---

> > > > > ### Author Response · Authors · 2023-11-16
> > > > >
> > > > > Thank you for raising the score.
> > > > >
> > > > > We will include those experiments in the final version.

---

> > > > > > ### Author Response · Authors · 2023-11-21
> > > > > >
> > > > > > We have updated **Table 15** with new results by considering your comment; please take a look into **Table 15** of the up-to-date revision. The results of **Table 15** are now matched to ones of **Table 1**.

---

> > > > > > > ### Comment · Reviewer_M9Vx · 2023-11-22
> > > > > > >
> > > > > > > I would like to thank the authors for the update.

---

### Official Review · Reviewer_rtfd · 2023-10-30

**Soundness:** 3 good
**Presentation:** 2 fair
**Contribution:** 2 fair
**Rating:** 6
**Confidence:** 3

**Summary:**

This paper examines an error-free differentiable swap function, which enables the creation of trainable sorting networks which do not accumulate errors as multiple applications of this function are performed. The authors evaluate the sorting network created using their swap function, and demonstrate improvements over previous differentiable sorting techniques.

**Strengths:**

- The proposed method is simple and easy to understand. The function used is equivalent to using the soft version of the maximum / minimum functions during the backward step, and the exact functions during the forward step. This allows for differentiability, without accumulating errors.  I especially appreciated the presentation of the samples in Figure 3, as it helped me understand how the method precisely works.

- The authors perform experiments on a variety of architectural choices and sequence lengths, which affect the accuracy of sorting algorithms. They show how incorporating an attention based part in the architecture is the better choice for sorting large sequences of data, since it provides a way to incorporate the entire sequence of data to sort and contextualize the output for each sample.

**Weaknesses:**

- Overall, while interesting, the applicability of the method seems limited at first glance, since one can in principle learn the function that maps samples to ordered objects first and then perform the sort using standard methods. I believe that the authors should either better motivate the need for this sort of method, or demonstrate in their experiments why learning the ordinal value directly does not perform as well.

- I think there is a gap in the proof of Proposition 2. Namely, the fact that the soft minimum and the soft maximum are the same as $k \to \infty$ seems like it is missing a step in how it is derived. I believe that it should hold, given the differentiability of the function, but filling in the missing step here is needed.

- I believe that certain parts can be improved with respect to the clarity of the paper. More specifically:

  - It seems to me that the paper assumes that the function used to soften the maximum is the sigmoid, and some of its properties are listed in Section 2 - however, it is not clear if the sigmoid is the only choice, or if these properties are sufficient for a function to satisfy. I would appreciate it if the authors could clarify this point.

  - Again in Section 2, I had some trouble understanding the wire-based construction of the permutation matrix, especially since the associated Figure is in the Appendix. I believe that the point of the authors would be clearer if they moved the Figure in the main paper, as well as simplify the explanation for the construction (saying, e.g., that each wire is a step to fix a single wrong ordering, and iterative application of these steps results in a sorted set).

  - I believe that Figures 1 and 2 can be simplified / combined, since they are essentially showing similar things - namely, that the function is error-free, which is not surprising given that it is not softened during the forward pass.

  - In Section 7, the paragraph “Utilization of Hard Permutation Matrices” is not very clear to me. I would appreciate it if the authors could rephrase the point they are trying to make here.

- As a minor point, in Section 6, the authors should rephrase how they refer to previous works, for example: “The work (Petersen et al., 2021) has suggested …” -> “Petersen et al. (2021) have suggested…”.

Overall, I think this is an interesting work, but I believe that the authors should improve upon the clarity of the paper and the motivation of their technique.

**Questions:**

As mentioned above, I would be grateful if the authors could clarify the paragraph “Utilization of Hard Permutation Matrices” of Section 7.

---

> ### Author Response · Authors · 2023-11-15
> **Response to Reviewer rtfd (1/n)**
>
> We appreciate your constructive comment to improve our work.
>
> > I believe that the authors should either better motivate the need for this sort of method, or demonstrate in their experiments why learning the ordinal value directly does not perform as well.
>
> By considering your comment, we have revised **Section N** of our work. For your convenience, we provide the updated paragraph that can enhance the motivation of our work here:
>
> It is challenging to directly sort a sequence of generic data instances without using auxiliary networks and explicit supervision. Unlike earlier sorting methods, this sorting network-based research (Petersen et al., 2021; 2022) including our work ensures that we can train a neural network that predicts numerical scores and eventually sorts them, even though we do not necessitate accessing explicit supervision such as exact numerical values of the contents in high-dimensional data. In this sense, the practical significance of our proposed methods can be highlighted by offering this possibility of solving a sorting problem with high-dimensional inputs. For example, as shown in Section 5, we can compare images of street view house numbers using the sorting network where our neural network is trained without exact house numbers.
>
> Moreover, instead of using costly supervision, our networks allow us to sort high-dimensional instances in a sequence where information on comparisons between instances is only given. This scenario often occurs when we cannot obtain complete supervision. For example, if we would sort four-digit MNIST images, ordinary neural networks are designed to solve a classification task by predicting class probabilities each of which indicates one of all labels from "0000" to "9999". If some labels are missing and further we do not know the exact number of labels, they might fail in predicting unseen data corresponding to those labels. Unlike these methods, it is possible to solve sorting problems with our networks in such a scenario.
>
> As discussed in Section 7, the hard permutation matrices produced by our methods encourage us to swap instances exactly, instead of the linear combination of instances. This characteristic is required when we are given the final outcomes of sorting as supervision. This scenario is tested by the experiments presented in Section 5.2. In these experiments, we can consider that original images are provided as supervision. Building on the advantages of neural network-based sorting networks, we extend their practical significance into the cases that need hard permutation matrices.
>
> Furthermore, this study can be applied in diverse deep learning tasks for learning to sort generic high-dimensional data, such as information retrieval (Cao et al., 2007; Liu, 2009) and top-$k$ classification (Berrada et al., 2018).

---

> ### Author Response · Authors · 2023-11-15
> **Response to Reviewer rtfd (2/n)**
>
> > I think there is a gap in the proof of Proposition 2. Namely, the fact that the soft minimum and the soft maximum are the same as $k \to \infty$ seems like it is missing a step in how it is derived. I believe that it should hold, given the differentiability of the function, but filling in the missing step here is needed.
>
> Thank you for pointing this out. We have updated the proof of Proposition 2. Please see **Section E**.
>
> > It seems to me that the paper assumes that the function used to soften the maximum is the sigmoid, and some of its properties are listed in Section 2 - however, it is not clear if the sigmoid is the only choice, or if these properties are sufficient for a function to satisfy. I would appreciate it if the authors could clarify this point.
>
> As described in **Section 2**, the required properties of $\sigma$ are that $\sigma(x)$ is differentiable, $\sigma(x)$ is non-decreasing, $\sigma(x) = 1$ if $x \to \infty$, $\sigma(x) = 0$ if $x \to -\infty$, $\sigma(0) = 0.5$, and $\sigma(x) = 1 - \sigma(-x)$.
> If a function is (point) symmetric, bounded, and differentiable by satisfying the aforementioned properties, that function should be $s$-shaped or sigmoid. Note that the sigmoid function in machine learning, which is widely used as an activation function, is called the logistic function in this paper.
>
> > Again in Section 2, I had some trouble understanding the wire-based construction of the permutation matrix, especially since the associated Figure is in the Appendix. I believe that the point of the authors would be clearer if they moved the Figure in the main paper, as well as simplify the explanation for the construction (saying, e.g., that each wire is a step to fix a single wrong ordering, and iterative application of these steps results in a sorted set).
>
> We have moved the figure and updated **Section 2**. Please refer to **Section 2** and **Figure 2**.
>
> > I believe that Figures 1 and 2 can be simplified / combined, since they are essentially showing similar things - namely, that the function is error-free, which is not surprising given that it is not softened during the forward pass.
>
> By considering your comment, **Figure 1** only remains in the main article. **Figure 2** is replaced with a figure with wire sets.
>
> > In Section 7, the paragraph "Utilization of Hard Permutation Matrices" is not very clear to me. I would appreciate it if the authors could rephrase the point they are trying to make here.
>
> > As mentioned above, I would be grateful if the authors could clarify the paragraph "Utilization of Hard Permutation Matrices" of Section 7.
>
> The experiments shown in **Section 5.2** solve a task to place image fragments on right positions. It requires swapping image fragments exactly, not combining image fragments via linear combination. Our sorting network ensures that final outcomes preserve initial instances from the application of sorting operations. We have also updated the corresponding paragraph of **Section 7**.
>
> > As a minor point, in Section 6, the authors should rephrase how they refer to previous works, for example: "The work (Petersen et al., 2021) has suggested ..." -> "Petersen et al. (2021) have suggested...".
>
> Thank you for pointing this out. We have fixed it and also similar cases; please see **Section 6**.

---

> ### Author Response · Authors · 2023-11-20
>
> We thank you for your constructive feedback again.
>
> We have answered your concerns in the rebuttal.
>
> Please take a look into our response and let us know if you still have any concerns.

---

> ### Comment · Reviewer_rtfd · 2023-11-20
> **Response to Author Rebuttal**
>
> I'd like to thank the authors for responding to my points. I now better understand the points in Section 2 and 7. For Section 7 in particular (as a minor point), I believe it would be good if the authors explicitly stated that $P_soft$ is not a good idea, precisely because it mixes representations, instead of simply sorting them (potentially replacing the current first sentence in the paragraph). I also now understand why simple supervision is not a good idea, since it is also affected by the amount of samples needed for each ordinal that we are trying to sort.
>
> Regarding the proof in Section E, I think it still contains a small issue - namely, the fact that $\bar{min}^k$ is strictly increasing and $\bar{max}^k$ is strictly decreasing does not (by itself) prove that these converge to the same limit (since it could be the case that $\lim_{k \to \infty} \bar{max}^k > \lim_{k \to \infty} \bar{min}^k$ - they could converge to different points). To fully complete the proof, I think something like $\lim_{k \to \infty} (\bar{max}^k - \bar{min}^k) = 0$ is necessary, which I believe can be derived by differentiability/continuity of $\sigma$. Explicitly showing that they both converge to $(max(x,y) + min(x,y)) / 2 = (x+y)/2$ would also suffice. I understand that this is a minor point overall, but I believe it should be corrected.

---

> > ### Author Response · Authors · 2023-11-20
> >
> > Thank you for your valuable feedback.
> >
> > We have update our manuscript based on your comment; please refer to **Sections 7 and E**.

---

> ### Author Response · Authors · 2023-11-22
>
> The reviewer-author discussion period is ending soon.
>
> If your concerns are resolved, please update your review.

---

### Official Review · Reviewer_Stfx · 2023-10-31

**Soundness:** 3 good
**Presentation:** 3 good
**Contribution:** 3 good
**Rating:** 6
**Confidence:** 3

**Summary:**

The paper introduces a new approach to sorting through neural networks. The authors extend traditional sorting mechanisms to handle more abstract and high-dimensional data types like multi-digit images and image fragments. The core contribution is an "error-free differentiable swap function," which allows the network to learn a mapping from complex inputs to sorted sequences effectively. This work aims to broaden the applicability of sorting algorithms by making them more versatile for dealing with intricate data types.

**Strengths:**

- Novelty: The paper introduces a novel "error-free differentiable swap function" to generalize sorting algorithms for more complex data types.
- Mathematical Rigor: The paper employs rigorous mathematical formulations to define the sorting problem and its extension for high-dimensional data.
- Methodological Approach: The use of a permutation-equivariant Transformer network with multi-head attention is a strong methodological choice for capturing dependencies between elements in the sequence.
- Clarity and Organization: The paper is well-organized and clearly written, making it accessible to readers who are familiar with the domain.

**Weaknesses:**

- Insufficient Discussion on Limitations: While the paper mentions that its methods are limited to sorting algorithms, it doesn't provide a thorough discussion on other potential limitations, such as scalability or conditions under which the method may not work well.
- Hyperparameter Sensitivity: The paper discusses various hyperparameters like steepness and learning rate but does not provide a comprehensive analysis of their impact, which could be a concern for practical implementations.
- Complexity of the Model: The use of a permutation-equivariant Transformer network with multi-head attention could make the model computationally expensive, which is not addressed in the paper.
- Missed Opportunity for Theoretical Analysis: While the paper is empirically driven, a more in-depth theoretical analysis could strengthen the paper by providing bounds or guarantees on the performance of the proposed methods.

**Questions:**

How sensitive is the model's performance to the choice of hyperparameters like steepness and learning rate? Could the authors provide more insights into this?
Could the authors provide a more comprehensive discussion on the limitations of their approach, particularly in terms of scalability and conditions where the method may not be applicable?

---

> ### Author Response · Authors · 2023-11-15
> **Response to Reviewer Stfx (1/n)**
>
> We appreciate your constructive comment to improve our work.
>
> > Insufficient Discussion on Limitations: While the paper mentions that its methods are limited to sorting algorithms, it doesn't provide a thorough discussion on other potential limitations, such as scalability or conditions under which the method may not work well.
>
> > Could the authors provide a more comprehensive discussion on the limitations of their approach, particularly in terms of scalability and conditions where the method may not be applicable?
>
> By considering your comment, we have updated **Sections M and N**.  We present those sections here:
>
> ### M. Limitations
>
> While a sorting task is one of the most significant problems in computer science and mathematics (Cormen et al., 2022), our ideas, which are built on sorting networks (Knuth, 1998; Ajtai et al., 1983), can be limited to sorting algorithms. It implies that it is not easy to devise neural network-based approaches to solving general problems in computer science, e.g., combinatorial optimization, which are inspired by our ideas. Nevertheless, the use of our algorithms enables us to learn a neural network-based sorting network for high-dimensional data, and employ the neural network in the tasks on sorting high-dimensional data instances.
>
> In addition, while our proposed methods show the superior performance compared to the baseline methods, this line of research suffers from the performance degradation for longer sequences as shown in Tables 1, 2, and 3. More precisely, for longer sequences, the element-wise accuracy of our methods does not decline dramatically, but the sequence-wise accuracy of our methods drops due to the nature of sequences. Incorporating our contributions such as the error-free DSFs and the Transformer-based networks, we expect that the further progress of neural network-based sorting networks can be achieved. In particular, the consideration of more sophisticated neural networks with a huge number of parameters, which are capable of handling longer sequences, might help improve performance. This will be left for future work.
>
> Furthermore, our frameworks successfully learn relationships between high-dimensional data with ordinal contents as shown in Section 5. However, our methods are supposed to fail in sorting data without ordinal information; the elaborate discussion on this topic can also be found in Section 7. In order to sort more ambiguous high-dimensional data, we can combine our work with part-based or segmentation-based approaches.
>
> ### N. Broader Impacts
>
> It is challenging to directly sort a sequence of generic data instances without using auxiliary networks and explicit supervision. Unlike earlier sorting methods, this sorting network-based research (Petersen et al., 2021; 2022) including our work ensures that we can train a neural network that predicts numerical scores and eventually sorts them, even though we do not necessitate accessing explicit supervision such as exact numerical values of the contents in high-dimensional data. In this sense, the practical significance of our proposed methods can be highlighted by offering this possibility of solving a sorting problem with high-dimensional inputs. For example, as shown in Section 5, we can compare images of street view house numbers using the sorting network where our neural network is trained without exact house numbers.
>
> Moreover, instead of using costly supervision, our networks allow us to sort high-dimensional instances in a sequence where information on comparisons between instances is only given. This scenario often occurs when we cannot obtain complete supervision. For example, if we would sort four-digit MNIST images, ordinary neural networks are designed to solve a classification task by predicting class probabilities each of which indicates one of all labels from "0000" to "9999". If some labels are missing and further we do not know the exact number of labels, they might fail in predicting unseen data corresponding to those labels. Unlike these methods, it is possible to solve sorting problems with our networks in such a scenario.
>
> As discussed in Section 7, the hard permutation matrices produced by our methods encourage us to swap instances exactly, instead of the linear combination of instances. This characteristic is required when we are given the final outcomes of sorting as supervision. This scenario is tested by the experiments presented in Section 5.2. In these experiments, we can consider that original images are provided as supervision. Building on the advantages of neural network-based sorting networks, we extend their practical significance into the cases that need hard permutation matrices.
>
> Furthermore, this study can be applied in diverse deep learning tasks for learning to sort generic high-dimensional data, such as information retrieval (Cao et al., 2007; Liu, 2009) and top-$k$ classification (Berrada et al., 2018).

---

> ### Author Response · Authors · 2023-11-15
> **Response to Reviewer Stfx (2/n)**
>
> On the other hand, this nature of neural network-based sorting networks can yield a potential negative societal impact. If this line of research including our proposed approaches is employed to sort controversial high-dimensional data such as beauty and intelligence, it can be thought of as the unethical use cases of artificial intelligence.
>
> > Hyperparameter Sensitivity: The paper discusses various hyperparameters like steepness and learning rate but does not provide a comprehensive analysis of their impact, which could be a concern for practical implementations.
>
> > How sensitive is the model's performance to the choice of hyperparameters like steepness and learning rate? Could the authors provide more insights into this?
>
> We have revised **Section L**. Please refer to **Section L**.
>
> > Complexity of the Model: The use of a permutation-equivariant Transformer network with multi-head attention could make the model computationally expensive, which is not addressed in the paper.
>
> We have shown the complexity of our model empirically. As shown in **Tables 1, 2, and 3**, We reported FLOPs and the number of parameters. When we determined the size of our Transformer-based network, we have considered these two factors. Moreover, discussion on this issue is described in the **Effects of Multi-Head Attention in the Problem (3)** paragraph of **Section 7**; please refer to this paragraph.
>
> > Missed Opportunity for Theoretical Analysis: While the paper is empirically driven, a more in-depth theoretical analysis could strengthen the paper by providing bounds or guarantees on the performance of the proposed methods.
>
> We have proved **Propositions 1, 2, and 3** where their proofs are provided in **Sections D, E, and F**. By these propositions, our error-free DSF can provide more accurate signals to the neural networks we are training. However, as described in your comment, our work is to propose a new framework of neural network-based sorting networks, which can be considered as the empirically driven work. More rigorous guarantees on the performance of the proposed methods will be left for future work.

---

> ### Author Response · Authors · 2023-11-20
>
> We thank you for your constructive feedback again.
>
> We have answered your concerns in the rebuttal.
>
> Please take a look into our response and let us know if you still have any concerns.

---

> ### Author Response · Authors · 2023-11-22
>
> The reviewer-author discussion period is ending soon.
>
> If your concerns are resolved, please update your review.

---

### Official Review · Reviewer_LP9f · 2023-11-01

**Soundness:** 2 fair
**Presentation:** 3 good
**Contribution:** 2 fair
**Rating:** 6
**Confidence:** 1

**Summary:**

The main content of the paper is the proposal of a generalized neural sorting network with error-free differentiable swap functions. The authors introduce scaled dot-product and multi-head attention and provide proofs for several propositions related to their network. They also discuss the effects of multi-head attention in the problem they are addressing and present empirical studies on the performance gains achieved by their proposed methods.

**Strengths:**

1. Generalization: The proposed neural sorting networks with error-free differentiable swap functions provide a general framework for solving sorting tasks on various types of data, including multi-digit images and image fragments.

2. Performance Improvement: The experimental results demonstrate that the proposed methods outperform the baseline method in terms of sorting performance, as measured by accuracy metrics.

3. Numerical Analysis: The authors provide a thorough numerical analysis of the effects of their methods, compared to the baseline method. This analysis validates the effectiveness of the proposed methods in sorting tasks.

4. In terms of writing, it is explained quite clearly, and one can generally follow the author's train of thought.

**Weaknesses:**

1. Since the methods proposed in the paper are based on sorting networks, they are not applicable to solving general problems in computer science, such as combinatorial optimization. This implies that it may not be easy to devise a neural network-based approach for solving these types of problems using the ideas presented in this work.

2. In terms of the results, it can be observed that as the sequence length increases or the number of segments grows, the performance decreases significantly.

3. I find it somewhat challenging to grasp the practical significance of this work. I believe that this work lacks practical significance to some extent.

**Questions:**

1. I hope the author can provide a detailed explanation of the practical significance of this work.

2. Why does the performance drop significantly when the quantity increases?

---

> ### Author Response · Authors · 2023-11-15
> **Response to Reviewer LP9f (1/n)**
>
> We appreciate your constructive comment to improve our work.
>
> > Since the methods proposed in the paper are based on sorting networks, they are not applicable to solving general problems in computer science, such as combinatorial optimization. This implies that it may not be easy to devise a neural network-based approach for solving these types of problems using the ideas presented in this work.
>
> > I hope the author can provide a detailed explanation of the practical significance of this work.
>
> > I find it somewhat challenging to grasp the practical significance of this work. I believe that this work lacks practical significance to some extent.
>
> By considering your comment, we have updated **Section N**. For your convenience, we present part of that section here:
>
> It is challenging to directly sort a sequence of generic data instances without using auxiliary networks and explicit supervision. Unlike earlier sorting methods, this sorting network-based research (Petersen et al., 2021; 2022) including our work ensures that we can train a neural network that predicts numerical scores and eventually sorts them, even though we do not necessitate accessing explicit supervision such as exact numerical values of the contents in high-dimensional data. In this sense, the practical significance of our proposed methods can be highlighted by offering this possibility of solving a sorting problem with high-dimensional inputs. For example, as shown in Section 5, we can compare images of street view house numbers using the sorting network where our neural network is trained without exact house numbers.
>
> Moreover, instead of using costly supervision, our networks allow us to sort high-dimensional instances in a sequence where information on comparisons between instances is only given. This scenario often occurs when we cannot obtain complete supervision. For example, if we would sort four-digit MNIST images, ordinary neural networks are designed to solve a classification task by predicting class probabilities each of which indicates one of all labels from "0000" to "9999". If some labels are missing and further we do not know the exact number of labels, they might fail in predicting unseen data corresponding to those labels. Unlike these methods, it is possible to solve sorting problems with our networks in such a scenario.
>
> As discussed in Section 7, the hard permutation matrices produced by our methods encourage us to swap instances exactly, instead of the linear combination of instances. This characteristic is required when we are given the final outcomes of sorting as supervision. This scenario is tested by the experiments presented in Section 5.2. In these experiments, we can consider that original images are provided as supervision. Building on the advantages of neural network-based sorting networks, we extend their practical significance into the cases that need hard permutation matrices.
>
> Furthermore, this study can be applied in diverse deep learning tasks for learning to sort generic high-dimensional data, such as information retrieval (Cao et al., 2007; Liu, 2009) and top-$k$ classification (Berrada et al., 2018).

---

> ### Author Response · Authors · 2023-11-15
> **Response to Reviewer LP9f (2/n)**
>
> > In terms of the results, it can be observed that as the sequence length increases or the number of segments grows, the performance decreases significantly.
>
> > Why does the performance drop significantly when the quantity increases?
>
> Since our neural networks are trained from scratch using only supervision on permutation matrices, the cases with longer sequences are hard to train. In addition, this performance drop becomes more significant because the impact of the performance drop is magnified as a sequence length increases. In particular, as presented in the results of $\textrm{acc}\_{\textrm{ew}}$, an individual score $s_i$ is relatively accurate. If a sequence length increases, $\textrm{acc}\_{\textrm{em}}$ exponentially drops. For example, if $\textrm{acc}\_{\textrm{ew}} = x$, $\textrm{acc}\_{\textrm{em}}$ is approximately $x^n$ where $n$ is a sequence length.
>
> It is worth noting that the previous studies struggled to show better performance in the same benchmarks and our methods outperform those baseline methods; please see **Tables 1, 2, and 3**.

---

> ### Author Response · Authors · 2023-11-20
>
> We thank you for your constructive feedback again.
>
> We have answered your concerns in the rebuttal.
>
> Please take a look into our response and let us know if you still have any concerns.

---

> ### Comment · Reviewer_LP9f · 2023-11-23
> **Feedback to author's response**
>
> Thanks for the reply. My problem was solved. I hope to present these in more detail in the final version. I will raise my score.

---

### Author Response · Authors · 2023-11-15
**General Comment to All Reviewers**

We thank the reviewers for their constructive feedback.

In particular, *Reviewer LP9f* mentioned that our method has the following strengths, ***generalization***, ***performance improvement***, ***numerical analysis***, and ***writing***. *Reviewer Stfx* commented that our work has the strengths as follows, ***novelty***, ***mathematical rigor***, ***methodological approach***, and ***clarity and organization***. *Reviewer rtfd* mentioned that ***the proposed method is simple and easy to understand*** and ***the authors perform experiments on a variety of architectural choices and sequence lengths, which affect the accuracy of sorting algorithms***. *Reviewer M9Vx* commented that our work is ***simple method***, ***better performance on long sequences***, and ***motivated by analyzing existing methods***.

Based on the reviewers' comments and suggestions, we have made the following improvements to our paper:

* Revised Section 2 (Sorting Networks with Differentiable Swap Functions)
* Moved Figure 2 to the appendices
* Moved a figure with wire sets to Figure 2
* Revised the Utilization of Hard Permutation Matrices paragraph of Section 7 (Discussion)
* Revised Section B (Comparisons of Different Sorting Networks)
* Updated Section E (Proof of Proposition 2)
* Added Section G (Details of Permutation-Equivariant Networks with Multi-Head Attention) and Figure 7
* Revised Section L (Study on Steepness and Learning Rate)
* Revised Section M (Limitations)
* Revised Section N (Broader Impacts)
* Fixed minor issues

---

### Meta-Review · Area_Chair_MR5h · 2023-12-14

**Metareview:**

The paper proposes a new loss and operator for differentiable sorting. The key idea is to exploit swap functions that are better suited for scenarios when the operator is applied repeatedly.  The authors provide an extensive study of the theoretical properties of this approach, along with an architecture based on a permutation-equivariant Transformer for validating the approach on downstream tasks and domains.
the reviewers had some questions regarding the practical significance of the problem, and the empirical ablations related to lengths of sequences, and ablations, which the authors resolved satisfactorially.

**Justification For Why Not Higher Score:**

Reviewers had valid concerns on the lack of some empirical ablations and limited motivation practical utility.

**Justification For Why Not Lower Score:**

The paper presents a novel solution to an interesting problem, provides theoretical evidence to justify their solution, and validates it decently on the chosen domains. No variance amongst the reviewers for suggesting a weak accept.

---

### Decision · Program_Chairs · 2024-01-16

Accept (poster)